



# Closing the data gap: runoff prediction in fully ungauged settings using LSTM

Reyhaneh Hashemi[1], Pierre Javelle[2], Olivier Delestre[3,4], and Saman Razavi[5,6]

[1]School of Engineering Sciences, Aix–Marseille University, Aix-en-Provence, France
[2]INRAE, RECOVER Research Unit, Aix–Marseille University, Aix-en-Provence, France
[3]Côte d'Azur University, CNRS, LJAD, Nice, France
[4]Saint-Venant Hydraulics Laboratory, Ecole des Ponts ParisTech - EDF R&D, Chatou, France
[5]School of Environment and Sustainability, Department of Civil, Geological and Environmental Engineering, Global Institute for Water Security, University of Saskatchewan, Saskatchewan, Canada
[6]Institute for Water Futures, Australian National University, Canberra, Australia

**Correspondence:** Pierre Javelle (pierre.javelle@inrae.fr)

**Abstract.** Prediction in ungauged basins (PUB), where flow measurements are unavailable, is a critical need in hydrology and has been a focal point of extensive research efforts in this field over the past two decades. From the perspective of deep learning, PUB can be viewed as a scenario where the generalization capability of a pretrained neural network is employed to make predictions on samples that were not included in its training data set. This paper adopts this view and conducts genuine PUB using long short-term memory (LSTM) networks. Unlike PUB approaches based on $k$-fold training-test technique, where an arbitrary catchment $B$ is treated as gauged in $k-1$ rounds and as ungauged in one round, our approach ensures that the sample for which the PUB is conducted (the UNGAUGED sample) is completely independent from the sample used to previously train the LSTMs (the GAUGED sample). The UNGAUGED sample includes 379 catchments from five hydrological regimes: Uniform, Mediterranean, Oceanic, Nivo–Pluvial, and Nival. PUB predictions are conducted using LSTMs trained both at the regime level (using only gauged catchments within a specific regime) and at the national level (using all gauged catchments). For benchmarking the performance of LSTM in PUB, four regionalized variants of the GR4J conceptual model are considered: spatial proximity, multi-attribute proximity, regime proximity, and IQ-IP-Tmin proximity, where $IQ$, $IP$, and $T_{\min}$ are the indices defining the five hydrological regimes. To align with the study's fully ungauged context, the $IQ$ index, which is also an input feature for the LSTMs, and the regime classification, crucial for the REGIME LSTMs, are reproduced under ungauged conditions using a regime-informed neural network and an XGBoost multi-class classifier respectively. The results demonstrate the overall superior performance of NATIONAL LSTMs compared to REGIME LSTMs. Among the four regionalization approaches tested for GR4J, the IQ-IP-Tmin proximity approach proves to be the most effective when analyzed on a regime-wise basis. When comparing the best-performing LSTM with the best-performing GR4J model within each regime, LSTMs show superior performance in both the Nival and Mediterranean regimes.



## 1 Introduction

Both conceptual and neural network-based runoff models are parametric models: the model defines a function described by a free-parameter vector whose size is finite and fixed prior to any data are observed. The free parameters allow the model to enhance its predictive capability during calibration. The calibrated parameter set when used in conjunction with the model's structure and solution scheme, generates a runoff response that is more accurate than other combinations and closely aligns with the observed response (Jacomino and Fields, 1997). A neural network comprises two types of parameters: 1. internal parameters (i.e. weights and biases), which are internal to the model and learnable, meaning that they are optimized during training by the model itself; and 2. hyperparameters, which are external to the model, are not learnable, and must be set and optimized by the user before training via hyperparameter tuning.

The spatial variability of the free parameters of these models depends on the range of variation in surface and subsurface variables observed across different regions. In hydrology, there has always been a need for models with a spatial-hydrological robustness: models that can simulate hydrologic responses over broad spatio–climatic ranges. When these parameters are determined at a local scale, the model's understanding of surface and subsurface processes becomes too specific to a single catchment. As a result, local calibration fails to capture the commonalities shared by different spatial locations. In fact, in a local calibration or training scenario, the model is unable to distinguish between large-scale patterns and detailed local specificities. Logically, as the model is exposed to data from an increasing number of catchments, its understanding of global trends becomes clearer. As a result, in a multi-catchment (regional) calibration or training scenario, it is more advantageous for the optimization algorithm to comprehend these global trends: it is only by comprehending these global trends that the algorithm can be effectively optimized across all the catchments. With the interest of leveraging such large-scale trends derived in regional scenarios, regionalization approaches have been developed to predict runoff in ungauged catchments where flow measurements are unavailable.

The prediction of runoff in ungauged basins (PUB) is an essential requirement in hydrology due to several reasons. Firstly, accurate knowledge of runoff is crucial due to its significance in water resource management, flood forecasting, and ecosystem protection. Secondly, even in countries with well-established gauged networks, a significant number of catchments, either partially or entirely, lack streamflow measurements (Kratzert et al., 2019). Consequently, PUB has become a focal point of extensive research efforts in hydrology over the past two decades (Hrachowitz et al., 2013; Razavi and Coulibaly, 2013; Parajka et al., 2013; Guo et al., 2021). In the following, a review focusing on PUB studies most relevant to the work in this paper is provided.

### 1.1 Traditional approaches

In traditional hydrology, there exist various PUB methods that differ in terms of complexity and success. The simplest method for predicting runoff in an ungauged catchment is to scale the observed hydrograph from an upstream or downstream gauge by the catchment area, as long as the ungauged catchment of interest is not located too far upstream or downstream from the stream gauge (Beven, 2011).



Another straightforward PUB method is the two-step regression-based approach. In the first step, catchment-wise calibration (i.e., calibration tailored to each individual catchment) of model parameters is performed for a group of gauged catchments (donors), resulting in a set of calibrated parameters for each donor. In the next step, a set of catchment attributes representative of catchment characteristics is selected as independent variables. Then, the calibrated parameters obtained in the first step are used as the dependent variables and a multiple linear regression model is fitted. This fitted linear regression model is subsequently applied to the desired ungauged catchment where the model parameters are either unavailable or treated as such (target). Early PUB methods predominantly fall into this category, but they have encountered limited success due to two major issues. Firstly, the methodological basis of the approach, relying on linear regression, is unable to capture parameter interactions effectively (Kuczera and Mroczkowski, 1998). Secondly, the problem of equifinality further challenges the accuracy of predictions (McIntyre et al., 2005; Wagener and Wheater, 2006; Kim and Kaluarachchi, 2008). To tackle the challenge of equifinality, solutions such as the one-step or simultaneous regression method are proposed, where model parameters are calibrated concurrently with the fitting of the regressor (Hundecha and Bárdossy, 2004). This involves optimizing all objective functions simultaneously, encompassing both the calibration of donor catchments and the regression of model parameters. Achieving a parameter–predictor formulation that is both parsimonious and effective is a non-trivial task within this approach. Regardless of being one or two-step, these methods collectively form the first category of PUB methods: regression-based class.

The second major class is centered on the concept of pooling a cluster of donors (Beven, 2011). This class involves two key methodological modules: one for selecting donors and another to utilize donor information. Different methods have emerged during the PUB Decade and after based on the diverse choices made for each of these two modules.

Considering the first module (donor selection approach), all methods are predicated on a condition: the donors should exhibit hydrological commonalities with the target. This sets these methods apart from linear regression-based methods, which do not impose any requirement of hydrologic similarity when selecting donors. However, there is no universally agreed-upon strategy for defining or identifying hydrologic similarity. Some approaches rely on spatial proximity (Merz and Blöschl, 2004; Oudin et al., 2008; Randrianasolo et al., 2011), where catchments in close proximity are considered more hydrologically similar and thus more suitable to be donors. Others define hydrological similarity in terms of distance in the space of physical catchment attributes (Merz et al., 2006; Oudin et al., 2008, 2010). The spatial proximity approach is subject to two limitations. Firstly, it assumes that there are no substantial changes in meteorological and geophysical surface conditions between the target catchment and its adjacent donors (Reichl et al., 2009; Alebachew et al., 2014). Secondly, the effectiveness of this approach depends on the spatial resolution of the gauged network density (McIntyre et al., 2005; Oudin et al., 2008). If the network is too sparse, the efficiency of the method would be significantly diminished.

Regarding the second module (donor information utilization), there are two common options. The first option is to regionalize parameters obtained from all donors to obtain a regionalized parameter set, which is then transferred to the target catchment for making predictions (Oudin et al., 2008; Beck et al., 2016). The second option is to use the parameter sets of each donor individually, transferring them to the target to perform independent simulations, and subsequently regionalizing the obtained outputs from different donors. Classical techniques for parameter/output regionalization include averaging and weighted averaging (Oudin et al., 2008; Randrianasolo et al., 2011), as well as extrapolation using methods such as Kriging (Stein, 1999) or





inverse distance weighting (Merz and Blöschl, 2004; Samuel et al., 2011).

The last class of traditional PUB methods revolves around the notion of hydrologic signatures. Following the approach introduced by Yadav et al. (2007), this class involves selecting a set of hydrologic signatures and catchment attributes, serving as model-independent metrics. Subsequently, a relationship is established between them to predict signatures from catchment attributes within an uncertainty framework; ultimately leading to the regionalization of hydrologic signatures. This method provides expected ranges of hydrologic signatures for ungauged catchments and was later applied in some early post-PUB studies such as Shu and Ouarda (2012), Pinheiro and Naghettini (2013), and Costa et al. (2014).

## 1.2 Modern deep learning-based approaches

Deep learning techniques, especially when supplemented with process-based knowledge, have demonstrated significant promise in hydrological modeling (Razavi et al., 2022). In the context of deep learning, the concept of PUB can be viewed as a scenario where the generalization ability of a pretrained neural network is leveraged to make predictions for new data samples that were not included in its original training data set: the neural network is trained on multiple gauged catchments to capture large-scale patterns; this acquired knowledge is then transferred to the local scale of a single ungauged catchment where predictions are intended to be made. This knowledge encompasses the learned internal parameters (weights and biases) and the tuned hyperparameters. This knowledge transfer is based on two key assumptions: 1. the two gauged and ungauged spaces are independent and identically distributed (i.i.d. assumptions of statistical learning theory; Vapnik, 1999), and 2. the factors that explain the variations in the gauged space also explain the variations in the ungauged space.

Performing PUB using deep learning represents an exemplary case of evaluating generalization. Figure 1 provides a schematic illustration of this concept. It is crucial to highlight that the ability to generalize in the ungauged space differs from the generalization ability within the gauged space, as evaluated in the experiments presented in Hashemi et al. (2022). In the gauged space, generalization is assessed for each catchment using its test period not used during training. However, other periods (namely, training and validation) from the same catchment are used during the training phase. Thus, the model is not entirely unfamiliar with the catchment in question.

Previous studies have investigated the application of deep learning-based methods for PUB; however, the number of such studies remains limited, of which only three studies are directly pertinent to this paper: Kratzert et al. (2019), Choi et al. (2022), and Nogueira Filho et al. (2022).

Kratzert et al. (2019) employ a 12-fold validation technique to mimic the ungauged settings. Their study focuses on a sample of 531 catchments from the CAMELS data set (Newman et al., 2015; Addor et al., 2017) within the climatic context of the United States. The sample is divided into 12 nearly equal subsets. In each of the 12 training–test rounds, denoted as round $k$, long short-term memory (LSTM; Hochreiter, 1998) training is conducted using all catchments except those belonging to subset $k$, i.e. using almost 487 catchments. The catchments within subset $k$ ($\approx 44$ catchments) are used for testing the trained LSTM model, representing ungauged predictions. The predictions from the 12 rounds are subsequently combined to form predictions for the entire sample. They compare their results against local simulations using both the lumped conceptual sacramento soil moisture accounting model (SAC-SMA; Burnash et al., 1973), as well as the distributed process-based national water model





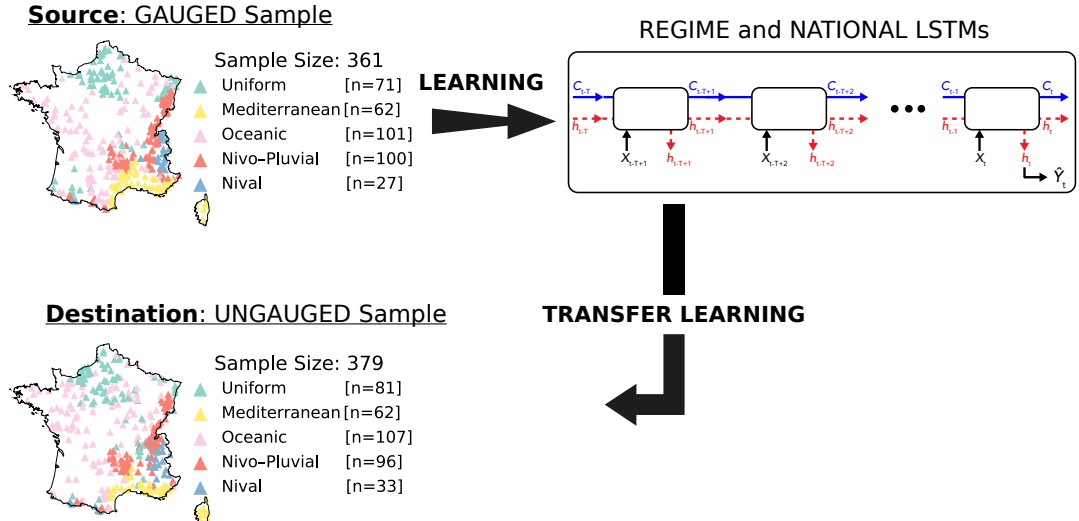

**Figure 1.** Applying deep learning for runoff prediction: bridging the gap from gauged to ungauged catchments. The catchments in both the top and bottom panels are classified by hydrological regimes, as defined by Hashemi et al. (2022). The top panel represents the gauged catchments used for model training, while the bottom panel illustrates the ungauged catchments where the trained model's generalization ability is tested across different hydrological scenarios. The color coding for hydrological regimes used in this figure is consistent with the rest of the paper to facilitate referencing of the specific regimes.

(NWM; Salas et al., 2018). Their findings indicate that, "on average (i.e., in more catchments than not)", their LSTM model outperforms both the SAC-SMA and NWM models in the ungauged settings.

Choi et al. (2022) follow the $k$-fold training–test framework of Kratzert et al. (2019) and use four LSTM models, differing only in their features, for 13 catchments in South Korea and observe promising results.

Nogueira Filho et al. (2022) use data from 25 catchments in the semiarid region of the State of Ceará, Brazil. They employ the leave-one-out training–test framework of Kratzert et al. (2019) to simulate the ungauged settings. Their study incorporates both a feedforward neural network and an LSTM network as deep learning runoff models. To evaluate their performance, a regionalized version (obtained by multiple linear regression) of the conceptual soil moisture accounting procedure (SMAP; Lopes et al., 1982) model is used as a benchmark. Their findings indicated the outperformance of their deep learning models in their study.

The $k$-fold validation based framework, initially employed by Kratzert et al. (2019) and subsequently adopted by the two other studies, exhibits the following characteristics:

- ungauged predictions are made using an out-of-bag (OOB) approach with a basis of bagging (Breiman, 1996), but with subsampling performed without replacement. This means that the method does not rely on a single learner with a fixed set of weights and biases. Instead, it involves 12 base learners, each trained with a different random subsample ($\approx 487$ catchments) selected from the total pool of catchments (531 in size). During the 12-shot training process, each





base learner generates OOB predictions for approximately 44 catchments that were not used in its training. Therefore, different ungauged catchments are not predicted using the same base learner. Indeed, the probability of two random catchments sharing the same learner depends on the value of $k$ and the sample size. For example, in Kratzert et al. (2019)'s study, this probability is found to be only 8% ($\approx \frac{43}{531-1}$).

- The OOB predictions do not replicate a genuinely ungauged setup, because in $k-1$ out of $k$ rounds the arbitrary catchment $B$ is treated as gauged, with existing data, while in 1 round it is treated as ungauged with no data. In a truly ungauged environment, there will always be a complete absence of data for an ungauged catchment, rendering OOB-type approaches infeasible.

- When multiple learners are used and their results are aggregated, the resulting performance represents an ensemble performance. This ensemble performance typically outperforms the performance of a single learner because the deficiencies of individual learners tend to average out when combined. However, in genuinely ungauged settings, where ensemble training data are unavailable, this advantage cannot be realized.

- The aggregated forms of the training and evaluation samples in the OOB-based framework are identical.

In this paper, the goal is to replicate a genuine ungauged scenario for PUB: catchment B is consistently treated as ungauged and is never included in any training tasks of the model used for making predictions. The experiments in this paper involve the catchment-wise application of REGIME and NATIONAL LSTMs (trained at both regime and national levels) in the previous study by Hashemi et al. (2022) to an entirely new sample, referred to as the UNGAUGED sample. These experiments enable a genuine evaluation of LSTM performance at both the regime and national levels. In the fully ungauged context of this study, the complete absence of discharge records presents additional challenges, specifically regarding the discharge index ($IQ$, $\frac{Q_{\max}-Q_{\min}}{Q_{\mean}}$; Hashemi et al., 2022). This index plays a dual role: first, it serves as an input feature for both the REGIME and NATIONAL LSTMs, and second, it is used in the regime classification that the REGIME LSTMs rely on, as they are exclusively applied to catchments that belong to their respective regimes.

To overcome these challenges, $IQ$ needs to be estimated separately. Here, multiple approaches for $IQ$ estimation are tested that incrementally increase in complexity and build upon each other. Once $IQ$ is estimated, the first challenge is already addressed. Now, it is necessary to determine the regime of the ungauged catchments to be able to pick their respective REGIME models (second challenge). To address this, the regime classification derived in Hashemi et al. (2022) is recreated in fully ungauged settings using a machine learning algorithm. Once the $IQ$ values are estimated, and the regime classification is reconstructed, the REGIME and NATIONAL LSTMs are applied catchment-wise to the UNGAUGED sample to generate two sets of ungauged predictions for each catchment. The predictions are conducted for the entire time series of each catchment since there is no need for data splitting in a fully ungauged space, which does not involve any LSTM training or hyperparameter tuning. Finally, the predictions made by LSTM are compared to the corresponding predictions made by several regionalized variants of the conceptual GR4J model (Perrin et al., 2003).

The data used in this paper comprise two samples: the GAUGED and UNGAUGED samples. The GAUGED sample is primarily used for calibration/training tasks, such as $IQ$ prediction and classification recreation, and can be further subdivided





into TRAINING and VALIDATION subsets based on the specific task and purpose. Conversely, the UNGAUGED sample is exclusively employed for evaluation or testing purposes.

The PUB study conducted in this paper contributes to the deep learning based PUB research in the following ways:

1. by conducting runoff predictions in "genuinely" ungauged settings;

2. by focusing on a significant sample of ungauged catchments within the French context, a scenario not previously explored;

3. by benchmarking the LSTM results with a conceptual model of a non-mass-conservative class, which accounts for water gains and losses within the catchment's boundaries. This comparison between LSTM and a non-mass-conservative
approach in the context of PUB is not found in the existing literature.

The remainder of this paper is structured as follows. Section 2 presents the data sets used. The methodological framework is described in Sect. 3. Section 4 elaborates the $IQ$ prediction approach. The recreation of the regime classification is described in Sect. 5. The regionalization of the GR4J model is conducted in Sect. 6. Results are provided in Sect. 7, which also discusses potential future research avenues based on the findings of this paper.

## 2   Data

This paper uses daily hydrometeorological time series of total precipitation ($P_{\text{tot}}$; mm day$^{-1}$); minimum ($TN$) and maximum ($TX$) air temperature (°C); wind speed ($WS$; m s$^{-1}$), specific air humidity ($HU$; g kg$^{-1}$), atmospheric radiation ($AR$; joule cm$^2$), visible radiation ($VR$; joule cm$^2$), precipitation solid fraction ($S_{\text{fr}}$), and potential evapotranspiration ($PET$; mm day$^{-1}$; Oudin et al., 2005) of a total of 740 catchments, sourced from the HydroClim database (Delaigue et al., 2020).
These catchments are from diverse geographical locations across France, each having a discharge time series ranging from 21 to 50 years. The spatial distribution of these catchments, along with their respective median elevations, is shown in Fig. 2. The time-invariant physical and hydroclimatic attributes used in this paper include median slope ($S$, %), drainage density ($DD$, %), area ($A$, km$^2$), median elevation ($Z_{50}$, m), discharge index ($IQ$; $\frac{Q_{\max}-Q_{\min}}{Q_{\text{mean}}}$; Hashemi et al., 2022), total precipitation index ($IP$; $\frac{P_{\max}-P_{\min}}{P_{\text{mean}}}$; Hashemi et al., 2022), min monthly temperature ($T_{\min}$; °C), mean daily liquid precipitation ($P_{\text{liq}}$;
mm day$^{-1}$), mean daily solid precipitation ($P_{\text{sol}}$; mm day$^{-1}$), and mean daily potential evapotranspiration ($PET$, mm day$^{-1}$). Figure 3 shows the spatial distribution of the hydroclimatic attributes and Table 1 presents the minimum, maximum, mean, and median values of each of the six climatic attributes and four physical descriptors for the 740 catchments.

### 2.1   GAUGED and UNGAUGED Samples

Within this 740-catchment sample, there are 391 catchments with a complete discharge time series spanning 30 years or more,
while the remaining 349 catchments have discharge time series of less than 30 years. From the subset of catchments with a discharge time series of 30 years or more, a sample of 361 catchments unaffected or minimally influenced by reservoirs and



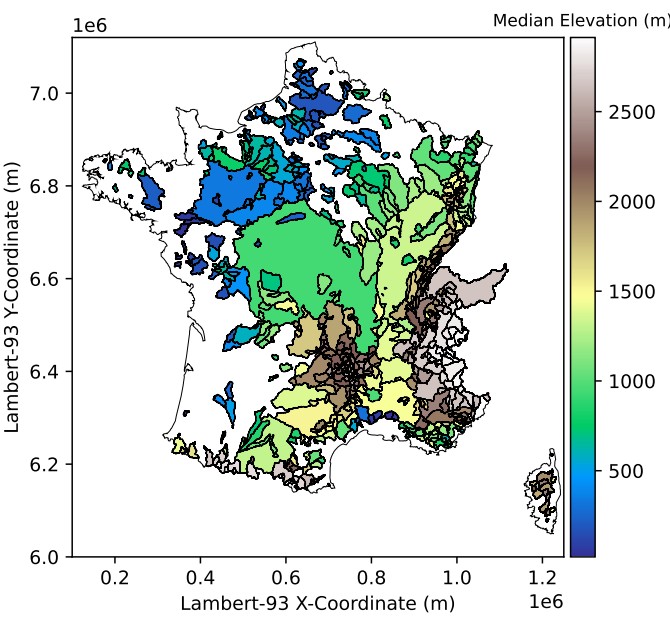

**Figure 2.** Geographical distribution of catchments in the NATIONAL sample

**Table 1.** Descriptive statistics of physical and hydroclimatic attributes for catchments in the NATIONAL sample

| Catchment characteristic | Notation | Min | Max | Mean | Median | SD[*] |
|---|---|---|---|---|---|---|
| Physical Attributes | | | | | | |
| Median Slope (%) | $S$ | 0 | 11 | 2 | 0 | 2 |
| Drainage Density (%) | $DD$ | 0 | 29 | 11 | 12 | 6 |
| Area (km$^2$) | $A$ | 5 | 13806 | 691 | 219 | 1494 |
| Median Elevation (m) | $Z_{50}$ | 20 | 2740 | 723 | 613 | 581 |
| Hydroclimatic Attributes | | | | | | |
| Runoff Index ($-$) | $IQ$ | 0.1 | 4.6 | 1.4 | 1.4 | 0.6 |
| Total Precipitation Index ($-$) | $IP$ | 0.2 | 1.7 | 0.6 | 0.5 | 0.4 |
| Minimum Monthly Temperature ($-$) | $T_{min}$ | -8.3 | 8.0 | 1.6 | 1.8 | 2.7 |
| Daily Liquid Precipitation (mm day$^{-1}$) | $P_{liq}$ | 0.9 | 5.1 | 2.6 | 2.5 | 0.7 |
| Daily Solid Precipitation (mm day$^{-1}$) | $P_{sol}$ | 0.0 | 3.3 | 0.4 | 0.2 | 0.5 |
| Daily Potential Evapotranspiration (mm day$^{-1}$) | $PET$ | 0.6 | 2.4 | 1.7 | 1.7 | 0.3 |

[*]SD: Standard Deviation

dams is identified. This sample is utilized for all calibration/training tasks and is referred to as the GAUGED sample.

The remaining 379 catchments are denoted as the UNGAUGED sample, which is used for ungauged scenarios. Within the

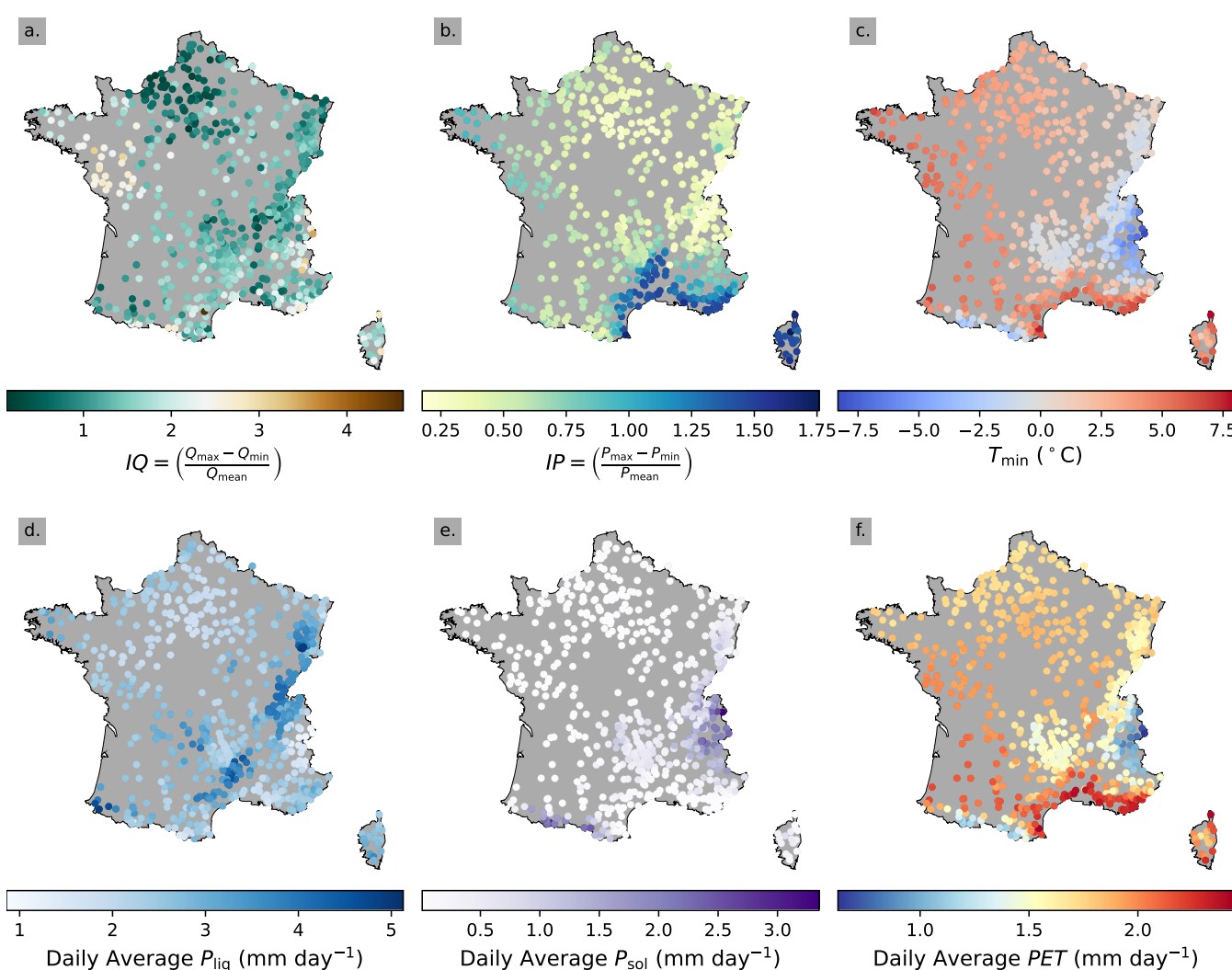

**Figure 3.** Spatial variations in a. runoff index ($IQ$), b. total precipitation index ($IP$), c. minimum monthly temperature ($T_{\min}$; °C), d. daily average liquid precipitation ($P_{\text{liq}}$; mm day$^{-1}$), e. daily average solid precipitation ($P_{\text{sol}}$; mm day$^{-1}$), and f. daily average potential evapotranspiration ($PET$; mm day$^{-1}$) across the 740 catchments of this study

UNGAUGED sample, 349 catchments possess complete discharge time series spanning more than 21 and less than 30 years,

while the remaining 30 have equal to or more than 30 years of full-record discharge data. The combination of varying data sizes and the presence of catchments with relatively small data sets is not a concern, as this sample is exclusively used for investigations related to ungauged simulations. In these scenarios, a pretrained model is applied to catchments not included in the training set, and no additional training takes place. The geographical distribution of the GAUGED and UNGAUGED samples is shown in Fig. 4 and Table 2 provides a comparison between the GAUGED and UNGAUGED samples, highlighting

the range of variation and median values for their key characteristics. The precise distribution of these variables is further





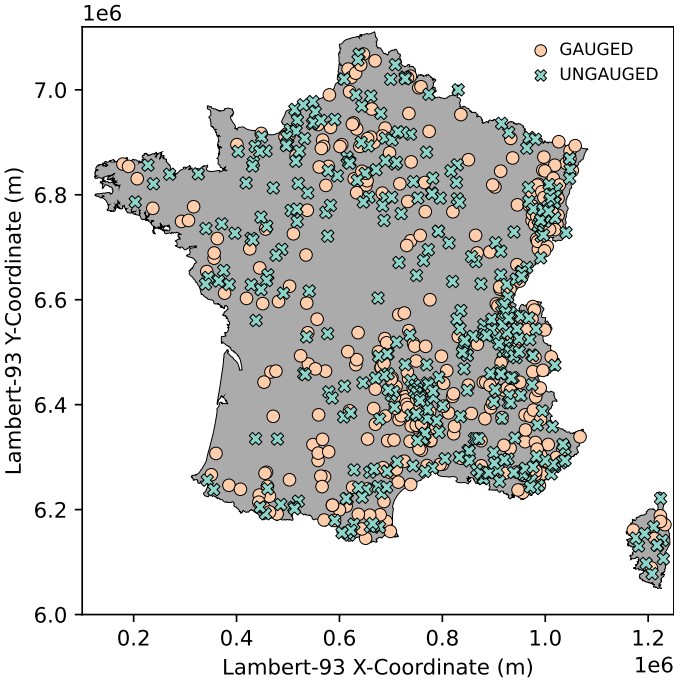

**Figure 4.** Geographical distribution of catchments within the GAUGED and UNGAUGED samples

illustrated in Fig. 5, demonstrating that the GAUGED and UNGAUGED samples primarily differ in terms of the length of their full-record discharges and they exhibit very similar distributions for all other variables.

## 3   Methodological framework

The methodological framework for conducting PUB in this study involves applying multi-catchment trained REGIME and
NATIONAL LSTM networks, as developed in Hashemi et al. (2022), to the 379 catchments in the UNGAUGED sample. This approach is designed to evaluate the models' performance in genuine PUB scenarios. The LSTM results are then compared with four regionalized variants of the GR4J model: spatial proximity, multi-attribute proximity, regime proximity, and IQ-IP-Tmin proximity (refer to Sect. 6). Each GR4J regionalized variant is derived using the traditional donor-target method, a common approach in hydrology for PUB. For both the LSTM and GR4J models, the $IQ$ variable is separately estimated in a fully
ungauged context using a regime-informed neural network (refer to Sect. 4). This variable serves as an input feature for both the REGIME and NATIONAL LSTMs and as an attribute in the multi-attribute and IQ-IP-Tmin proximity methods. Additionally, the regime classification, necessary for both the REGIME LSTMs and the GR4J regime proximity approach, is recreated in a fully ungauged context using an XGBoost multi-class classifier (refer to Sect. 5). A flowchart of this methodological framework is depicted in Fig. 6. The results and discussions pertaining to these methods are detailed in the subsequent sections.



**Table 2.** Comparison of key characteristic ranges and medians in GAUGED and UNGAUGED samples

| Attribute | | GAUGED | UNGAUGED |
|---|---|---|---|
| Median Slope ($S$; %) | Range | 0 to 10.8 | 0 to 9.3 |
| | Median | 0.4 | 0.3 |
| Area ($A$; km$^2$) | Range | 5 to 13806 | 2 to 96016 |
| | Median | 219 | 163 |
| Drainage Density ($DD$; %) | Range | 0 to 29 | 0 to 29 |
| | Median | 12 | 11 |
| Annual Average Runoff ($Q$; mm year$^{-1}$) | Range | 47 to 2312 | 25 to 1910 |
| | Median | 466 | 416 |
| Annual Average Total Precipitation ($P_{\mathrm{tot}}$; mm year$^{-1}$) | Range | 621 to 2128 | 563 to 2290 |
| | Median | 1053 | 995 |
| Annual Average Temperature ($T$; °C) | Range | -1.1 to 14.7 | -1.8 to 14.8 |
| | Median | 9.3 | 9.6 |

## 4 Prediction of $IQ$ variable in ungauged settings

The $IQ$ index provides information on the variability of runoff throughout the year. A low value for $IQ$ indicates a uniform distribution of runoff across the year, whereas a high value indicates the presence of contrasting dry and wet seasons (Hashemi et al., 2022). To fully align with the ungauged context, this variable is independently estimated to be used as an input feature for LSTM models.

Mathematically, estimating $IQ$ involves solving a multiple regression problem, where the objective is to establish the relationship between a set of independent variables (also referred to as predictors or features) and one dependent variable, $IQ$ (also known as the target variable in machine learning terminology). To accomplish this, a regression model is selected and fitted to data that contain both the predictors and the target variable. The fitted model can then be used to estimate the target variable for cases where only the predictors are available (e.g. ungauged scenarios).

Here, the GAUGED sample is used to fit several multivariate regression models with varying levels of complexity. These fitted models are subsequently employed to predict $IQ$ for the 379 catchments in the UNGAUGED sample. The selected predictors include the following variables: WGS X-Y coordinates (m), median altitude ($Z50$; m), median slope ($S$; %), drainage density ($DD$; %), as well as the two climatic classification variables used in Hashemi et al. (2022), namely the total precipitation index ($IP$; $\frac{P_{\max} - P_{\min}}{P_{\mean}}$) and minimum monthly temperature ($T_{\min}$; °C). The inclusion of X-Y coordinates accounts for the geograph-



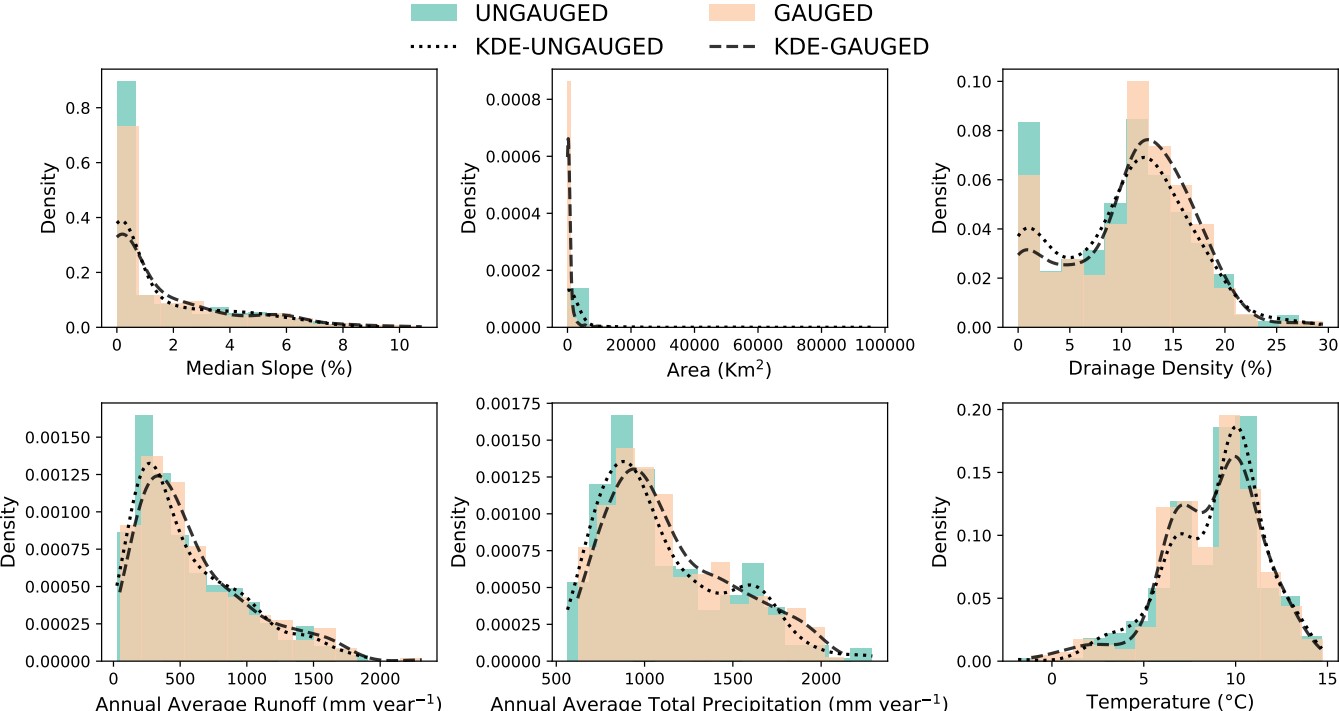

**Figure 5.** Histogram and kernel density estimation (KDE) for key catchment attributes in the GAUGED and UNGAUGED samples

ical location of different catchments, which can have a correlation with $IQ$ values below 1, indicating Uniform catchments that form distinct patches in the north and northeast regions of the country (see Fig. 3 in Hashemi et al., 2022). Drainage density, slope, and median altitude are included to capture the relationship between hydrology and geomorphology. Among these predictors, drainage density is expected to have a higher importance in relation to $IQ$, as it reflects the infiltration and permeability characteristics of a catchment: a Uniform catchment with a relatively low $IQ$ value, resulting from significant

aquifer impacts, would exhibit less efficient drainage compared to catchments without aquifer impacts.

The regression problem of predicting $IQ$ presents significant challenges in ungauged settings: the predictors cannot contain any explicit information about discharge. Due to this constraint, only predictors with an indirect relationship with discharge are allowed to be used. This makes it challenging to extract meaningful information from these predictors. Moreover, the IQ index itself is complex, being a composite variable constructed from three variables: minimum, maximum, and average flow.

Also, the number of samples available for this regression problem is only 361, which is a very limited data set, especially considering the complexity of the problem. As a result, finding an optimal regressor becomes a delicate balance between complexity and simplicity. The regressor should not be overly complex due to the limited data availability, while also avoiding oversimplification as the regression problem itself is inherently difficult.





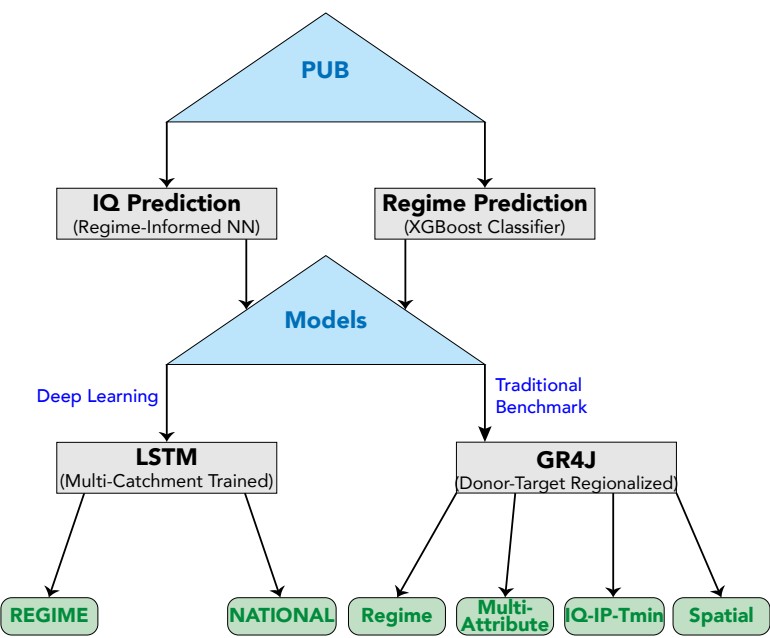

**Figure 6.** Schematic methodology for PUB in this paper using multi-catchment trained regime and national LSTMs benchmarked against regionalized GR4Js

## 4.1 Multiple linear regression (Benchmark)

The simplest form of regression tested in this study is multiple linear regression. It serves as a benchmark for all the more complex methods. It measures the degree of linear relationship between the target variable ($IQ$) and each predictor. Taking the seven predictors of the 361 catchments in the GAUGED sample, standardizing them, and fitting a multiple linear regression model, The fitted linear regressor is applied catchment-wise to the UNGAUGED sample and the corresponding results are shown in Fig. 7: the linear regression model yields an $R^2$ coefficient of 0.32 and 0.27, a mean absolute error (MAE) of 0.33 and

0.44, and a Kling–Gupta efficiency (KGE; Gupta et al. (2009)) of 0.39 and 0.30 for the GAUGED and UNGAUGED samples respectively.

## 4.2 Non-linear regression using neural networks

In the previous section, it was observed that the linear regressor was overly simplistic for the $IQ$ prediction task. This section will explore artificial neural networks as a non-linear alternative. To begin training, the training data (i.e., the GAUGED sample)

is divided into TRAINING and VALIDATION sets. The splitting is performed with a ratio of 0.8 for the training set and 0.2 for the validation set. The data is standardized using the mean and standard deviation calculated from the training set. Then, the hyperparameters and variations presented in Table 3 are considered. For each of the 144 ($= 2 \times 3 \times 2 \times 4 \times 3$) hyperparameter tuning cases presented in Table 3, the respective neural network is configured with linear layers followed by the application



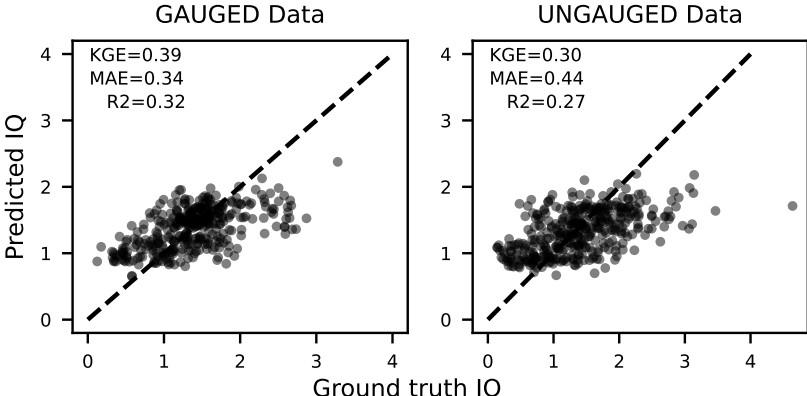

**Figure 7.** Calibration and evaluation performance scores (KGE, MAE, $R^2$) for $IQ$ prediction using multiple linear regression and seven predictors: X- and Y-coordinates, median altitude ($Z_{50}$), $IP$, $T_{\min}$, median slope ($S$), and drainage density ($DD$)

**Table 3.** Hyperparameter tuning for $IQ$ prediction using neural networks

| Hyperparameter | Considered Variations |
|---|---|
| Number of layers | 1, 2 |
| Number of hidden units (per hidden layer) | 32, 64, 128 |
| Learning rate | 0.001, 0.0001 |
| Batch size | 2, 4, 8, 16 |
| Dropout rate | 0, 0.1, 0.2 |

of a hyperbolic tangent activation function (except for the output layer), and uses a mean squared error (MSE) loss function. The network is trained for 1000 epochs on the TRAINING set and validated on the VALIDATION set and the KGE score is calculated at the end of each epoch. The weights and biases associated with the epoch that achieved the maximum validation KGE are chosen as the final model parameters. Upon comparing all 144 scores, the set of hyperparameters that achieved the best validation KGE (0.782) is as follows: number of hidden layers, 2; number of hidden units, 32 (per hidden layer); learning rate, 0.001; batch size, 8; and dropout rate, 0. The performance of this neural network is illustrated in Fig. 8. Compared to the multiple linear regression model, significant improvements in test performance are observed across all three metrics:

- KGE: $0.348 \rightarrow 0.644$,     • MAE: $0.437 \rightarrow 0.348$,     • $R^2$ coefficient: $0.294 \rightarrow 0.502$.

### 4.3 Non-linear regression using regime-informed neural networks

To enhance performance when using neural networks, a regularization technique is employed to integrate additional information about the catchment regime during training. This is achieved by imposing an additional condition as a penalty term to the loss function, ensuring its minimization while minimizing the discrepancy between the neural network $IQ$ predictions and the



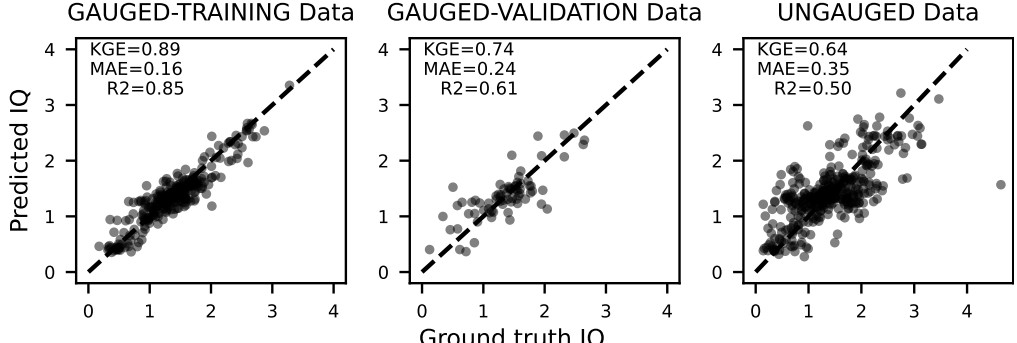

**Figure 8.** Performance scores (KGE, MAE, and R$^2$) on training, validation, and test data sets in $IQ$ regression using a neural network with two hidden layers

true $IQ$ values (ordinary loss term). The additional condition is established based on an important piece of information about the catchment regime, which is made available at the end of each forward pass.

In the context of the classification algorithm discussed in Hashemi et al. (2022), a classification rule relies on the $IQ$ value: whether it is lower than 1 or not. This condition can be evaluated at the end of each forward pass for both the true and predicted

$IQ$ values, and their discrepancy can be incorporated into the loss as a penalty term, hereafter referred to as the regime-informed loss. By minimizing the total loss, the network is encouraged to minimize both the error in:

1. the thresholding of the $IQ$ values, as indicated by the regime-informed loss, i.e., the discrepancy between $1 < (IQ)_{\text{true}}$ and $1 < (IQ)_{\text{predicted}}$; and,

2. the absolute $IQ$ values, as indicated by the ordinary loss, i.e., the discrepancy between $(IQ)_{\text{true}}$ and $(IQ)_{\text{predicted}}$.

The calculation process for the regime-informed loss is as follows. After the forward pass, the relative positions of the true and predicted $IQ$ values with respect to the value 1 is calculated: whether $IQ - 1 < 0$. This results in two boolean vectors: one obtained from the true values and one from the predicted values. In an ideal scenario, these vectors should be identical, indicating a perfect alignment between the positions of predicted $IQ$ values and true $IQ$ values with respect to the value 1. However, if there are disparities between the two vectors, it suggests that some Uniform catchments are being misclassified

as non-Uniform, and some non-Uniform catchments receive predicted $IQ$ values falling within the range of Uniform catchments. To quantify these discrepancies, a comparison is made between each corresponding pair of values in the two boolean vectors and the number of cases that exhibit a discrepancy is counted ($N_{\text{missclassified}}$). For such instances, the regime-informed loss is obtained by calculating their $L_1$-norm lasso (least absolute shrinkage and selection operator; Tibshirani (1996)) loss: $||(IQ)_{\text{true}} - (IQ)_{\text{predicted}}||_{L_1}$. Finally, the regime-informed loss is scaled by the number of cases for which it has been calcu-

lated in order to balance it with the ordinary MSE loss. The scaled loss is then added to the ordinary MSE loss as a regularization





term with a weight $\beta$:

$$l_{\text{total}} = \underbrace{\text{MSE}\big((IQ)_{\text{true}}, (IQ)_{\text{predicted}}\big)}_{\substack{\text{Ordinary Loss} \\ \text{Calculated on all points in the batch}}} + \underbrace{\beta||(IQ)_{\text{true}} - (IQ)_{\text{predicted}}||_{L_1} \times N_{\text{missclassified}}^{-1}}_{\substack{\text{Regime-informed Loss} \\ \text{Calculated on only misclassified points in the batch}}}. \tag{1}$$

The $\beta$ weight, serving as a hyperparameter, can be tuned to adjust the importance of the regime-informed loss relative to the original loss.

The regime-informed neural network described above, using the number of layers, hidden units, learning rate, batch size, and dropout rate determined in Subsect. 4.2, undergoes tuning for the parameter $\beta$ across five values: 0.05, 0.08, 0.1, 0.2, 0.3. Among these, a value of 0.1 achieves the highest validation KGE score, as detailed in Table 4.

Figure 9 shows the training and validation loss evolution of this neural network. The loss curves obtained from the network optimized without regime-based regularization are also juxtaposed for comparison. It is observed that adding the regime-

**Table 4.** Validation KGE scores while tuning regime-informed loss weight in the neural network used for $IQ$ prediction

| Weight of $IQ$-condition loss | Validation KGE score |
| :---: | :---: |
| 0.05 | 0.740 |
| 0.08 | 0.740 |
| 0.1 | 0.770 |
| 0.2 | 0.766 |
| 0.3 | 0.760 |

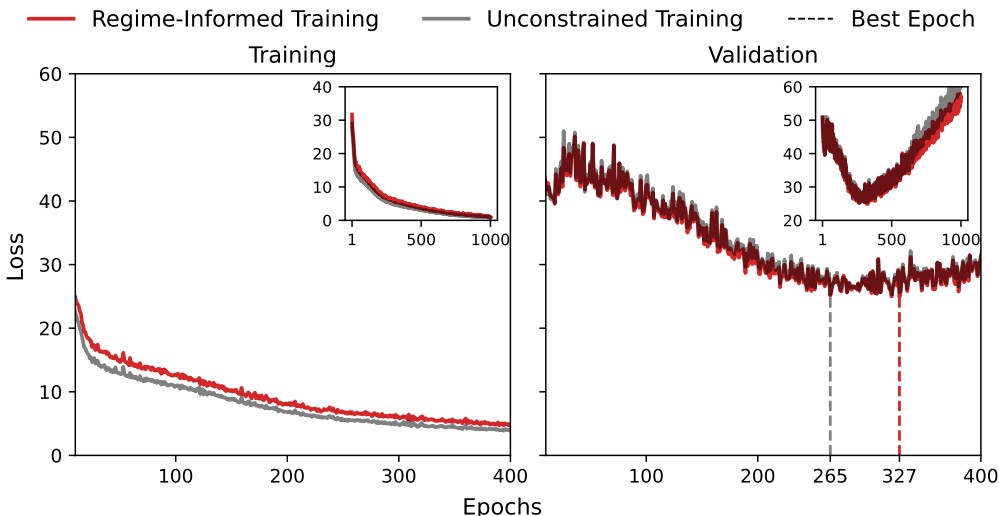

**Figure 9.** Learning curve for the regime-informed neural network used for $IQ$ estimation

informed loss term increases the training loss while decreases the validation loss; ultimately improving model performance on

the test data, albeit not substantially.

Indeed, regularization terms are known to explicitly reduce generalization error, often at the cost of an increase in training

error (Goodfellow et al., 2016). The limited significance of the observed improvement in generalization here may be attributed

to the notably small batch size, which reduces the likelihood of encountering cases with incorrect $IQ$ thresholding: in some

forward passes, there might even be none of them. Moreover, the learning curve of the constrained network exhibits a slower

convergence rate, with the optimal epoch occurring at 327 compared to 265 for the unconstrained network. This behavior is

consistent since the penalty term discourages the model parameters from becoming excessively large, which can help prevent

overfitting but may also decelerate the learning process. Using this network with a $\beta = 0.1$ configuration, the results shown in

Fig. 10 are obtained for the TRAINING, VALIDATION, and test (i.e. UNGAUGED sample) data.

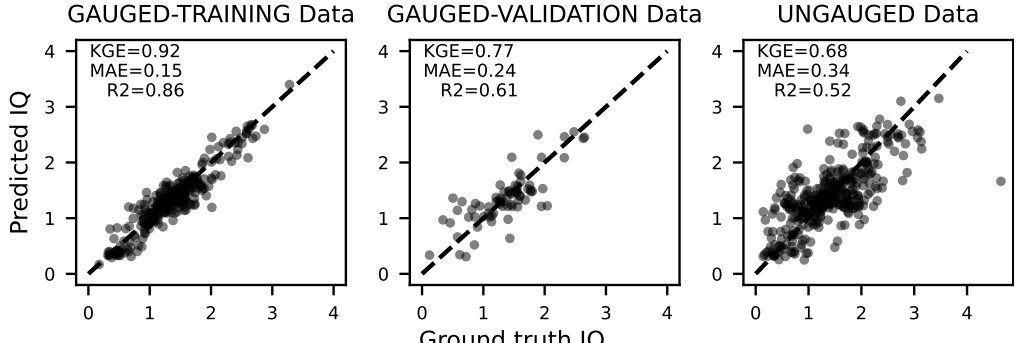

**Figure 10.** Performance scores (KGE, MAE, $R^2$) on training, validation, and test data sets in IQ regression using the regime-informed neural network

Compared to the unconstrained neural network, improvements in KGE scores are observed across all data sets:

• TRAINING: $0.886 \rightarrow 0.924$, • VALIDATION: $0.738 \rightarrow 0.773$, • Test: $0.642 \rightarrow 0.684$.

The regime-constrained neural network described above (with $\beta = 0.1$) presents the best test performance, and its $IQ$ predictions are used hereafter for conducting PUB.



## 5 Recreating regime classification in ungauged settings

### 5.1 Method

Hashemi et al. (2022) proposed the following regime classification:

| | |
|---|---|
| Nival: | $T_{\min} \leq -2\,°\mathrm{C}$, |
| Nivo–Pluvial: | $-2 < T_{\min} < 0$, |
| Mediterranean: | $T_{\min} \geq 0$ and $IP > 1$, |
| Uniform: | $T_{\min} \geq 0$ and $IP \leq 1$ and $IQ < 1$, |
| Oceanic: | $T_{\min} \geq 0$ and $IP \leq 1$ and $IQ \geq 1$. |

In a fully ungauged scenario, this classification encounters limitations due to its reliance on the $IQ$ variable. While the Nival, Nivo–Pluvial, and Mediterranean regimes can be identified based on the $T_{\min}$ and $IP$ conditions, distinguishing between the Uniform and Oceanic regimes depends exclusively on the $IQ < 1$ condition. Consequently, classifying these two regimes becomes unfeasible in ungauged scenarios. To address this limitation, two potential solutions can be considered:

1. use a classifier trained on the GAUGED sample, where $IQ$ values are available, and then apply it to the UNGAUGED sample;

2. use the $IQ$ values obtained in Subsect. 4.3 and employ the existing classification method.

Here both approaches are evaluated, and the one that yields the best results is selected for regime classification in the UNGAUGED catchments.

### 5.1.1 Classification recreation using the XGBoost classifier

The goal is to train a classifier that replicates the classification used to assign regimes to the catchments within the GAUGED sample, on which the REGIME LSTMs rely. Given that both the GAUGED sample and the corresponding regimes are accessible, this task aligns with supervised classification in machine learning.

In this context, extreme gradient boosting (XGBoost; Chen and Guestrin (2016)) is selected as the classifier for this task. The choice of XGBoost as the classifier is rooted in the fact that the target classification follows a tree-like logical trajectory characterized by if-then-else conditions. The selection of a classifier with a similar logical framework is therefore warranted. XGBoost is considered one of the top-performing tree-based algorithms, alongside random forest (RF; Breiman (2001)). However, in the context of the present classification task, XGBoost has been chosen.

The predictive variables selected for the XGBoost classifier comprise:

– average annual total precipitation ($P_{\mathrm{tot}}$; mm year[-1]),

– average annual evapotranspiration calculated using the formula from Oudin et al. (2005) ($PET$; mm year[-1]),

– the seven predictors used in predicting $IQ$: WGS X-Y coordinates, median slope ($S$; %), median altitude ($Z_{50}$; m), drainage density ($DD$; %), $IP$, and $T_{\min}$ (°C).





XGBoost offers a diverse set of hyperparameters that can be adjusted to influence the behavior of the algorithm. Here, the primary focus is on tuning the main hyperparameters (Chen and Guestrin, 2016), while keeping all other parameters at their default values (as implemented in the XGBoost Python package). The considered hyperparameters and their respective variations are as follows:

- maximum depth of a tree: 1, 3, 5, 7, 9;

- learning rate: 0.01, 0.1, 0.001;

- number of rounds for boosting: 5, 10, 15, 20.

To tune theses hyperparameters, the GAUGED sample is first divided into two sets: a training set and a validation set, with proportions of 0.8 and 0.2 respectively. The XGBoost classifier is then trained using the training data for each hyperparameter
set, and its respective performance is evaluated on the validation data. Accuracy scores for both the training and validation phases are recorded for each hyperparameter set. The hyperparameter set with the highest validation accuracy is then selected, and it is found to have the following values:

- max depth of tree: 5;

- learning rate: 0.1;

- number of rounds: 15.

Using these values, the XGBoost classifier is trained on the entire GAUGED data set, without splitting it into training and validation sets; resulting in an accuracy of 0.98. Subsequently, the trained model is employed to predict the regimes of UN-GAUGED catchments.

## 5.2 Results

Besides employing XGBoost as the classifier, the regime classification is also carried out using the classification rules outlined in Hashemi et al. (2022) and the predicted IQ values. The same accuracy level of 0.90 for the UNGAUGED sample is obtained by both approaches. This means the proportion of correctly predicted regimes that exactly matches the true regimes is the same in both methods. However, they do not result in the same regime distributions. Tables 5 and 6 present the contingency matrices for the two approaches.

In the contingency tables, the actual count of regimes (determined by the $IP$, $IQ$, $T_{\min}$ criteria and the actual $IQ$ values) is labeled under "true", while the count of regimes assigned by the classification approaches is listed under "predicted". TP denotes true positives, signifying instances that belong to a class and are correctly classified as such. FP corresponds to false positives, indicating instances that do not belong to a class but are incorrectly classified as belonging to it. FN represents false negatives, which are cases that belong to a class but are not classified as such.

It is observed that the XGBoost classifier successfully learns the rules related to $IP$ and $T_{\min}$, as the total number of true



**Table 5.** Contingency table for multi-class regime classification of catchments in the UNGAUGED sample using the XGBoost classifier

| Regime | Number of | | | | | Distribution of FNs |
| | True | Predicted | TP | FP | FN | |
| --- | --- | --- | --- | --- | --- | --- |
| Uniform | 81 | 66 | 56 | 10 | 25 | 23×Oceanic + 1×Nivo–Pluvial + 1×Mediterranean |
| Oceanic | 107 | 118 | 95 | 23 | 12 | 10×Uniform + 2×Mediterranean |
| Mediterranean | 62 | 65 | 62 | 3 | 0 | — |
| Nivo–Pluvial | 96 | 97 | 96 | 1 | 0 | — |
| Nival | 33 | 33 | 33 | 0 | 0 | — |

**Table 6.** Contingency table for multi-class regime classification of UNGAUGED catchments using the classification rules derived by Hashemi et al. (2022) and predicted $IQ$ values

| Regime | Number of | | | | | Distribution of FNs |
| | True | Predicted | TP | FP | FN | |
| --- | --- | --- | --- | --- | --- | --- |
| Uniform | 81 | 60 | 52 | 8 | 29 | 29×Oceanic |
| Oceanic | 107 | 128 | 99 | 29 | 8 | 8×Uniform |
| Mediterranean | 62 | 62 | 62 | 0 | 0 | — |
| Nivo–Pluvial | 96 | 96 | 96 | 0 | 0 | — |
| Nival | 33 | 33 | 33 | 0 | 0 | — |

positives (TPs) in the Mediterranean, Nival, and Nivo–Pluvial regimes match their actual total count. However, due to the absence of the $IQ$ variable, the classifier encounters difficulties in distinguishing between the Uniform and Oceanic regimes. This challenge arises because the actual classification of these two regimes relies solely on an $IQ$ condition (whether it is smaller than one or not), which cannot be evaluated in this context.

Comparing the two approaches, all catchments in the Mediterranean, Nival, and Nivo–Pluvial regimes are correctly classified by both methods. However, the XGBoost approach exhibits higher accuracy in the Uniform regime, while the other approach performs better in the Oceanic regime. Despite both approaches yielding the same overall accuracy, the decision is made here to select the XGBoost classifier due to its better performance in the Uniform regime. This regime has specific sequence length requirements, and it is essential to avoid using an unmatched REGIME LSTM for its catchments as much as possible.

The matches and discrepancies between the classification produced by XGBoost and the target classification are illustrated in Fig. 11 across various regimes within the UNGAUGED catchments.



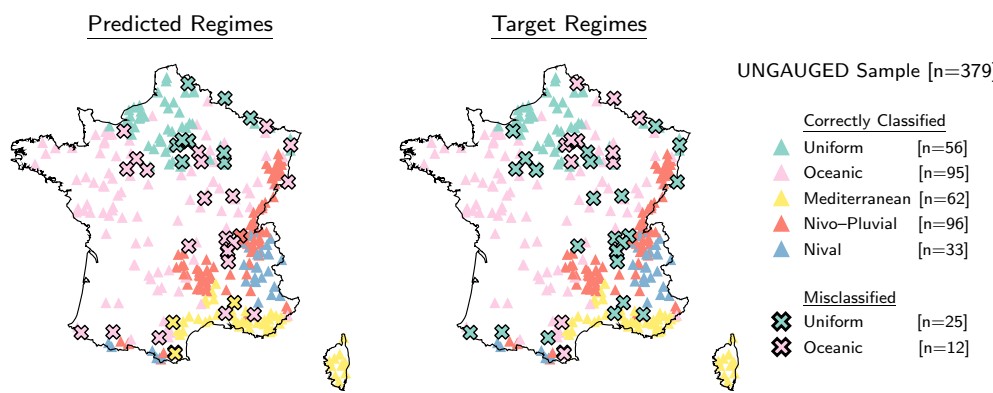

**Figure 11.** Comparison of true regime classes and XGBoost-predicted regimes for catchments in the UNGAUGED sample

# 6 Conceptual benchmarking using GR4J

## 6.1 Submethod 1: spatial proximity

The spatial proximity approach is predicated on the assumption that reducing the spatial distance between two catchments leads to greater hydrologic similarity. This is because as the spatial distance decreases, the hydro-climatic conditions and factors influenced by geographical features, which can impact catchment response, become more comparable (Oudin et al., 2008). Spatial proximity of the two arbitrary catchments $i$ and $j$ is defined as the Euclidean distance between their centroids in a two-dimensional space:

$$d_{\mathrm{spatial}}(i,j) = \sqrt{(x_{i,\mathrm{cen}} - x_{j,\mathrm{cen}})^2 + (y_{i,\mathrm{cen}} - y_{j,\mathrm{cen}})^2}, \tag{2}$$

where $x$ and $y$ denote the X and Y coordinates in the Lambert-93 coordinate system. The subscript "cen" refers to the centroid of each catchment. As Equation 2 suggests, when the spatial distance between a target catchment and itself is calculated, the resulting distance is 0. This indicates that the target catchment can serve as its own best donor, resulting in the local calibration of model parameters specifically for the target catchment.

In this approach, the number of donors is a crucial parameter that needs calibration. To do this while adhering to the fully ungauged framework, the following procedure is employed. The 361 catchments within the GAUGED sample are considered as the donors. For each donor, its available data is divided into two parts: the calibration set and the evaluation set. The evaluation set consists of the most recent 10 years of complete discharge records, while the remaining data form the calibration set. The coupled GR4J model is then calibrated for each donor using its calibration set, resulting in 361 sets of calibrated parameters $(X1, \cdots, X6)_1$ to $(X1, \cdots, X6)_{361}$. This process is referred to as "donor calibration". Next, a range of values is considered as potential numbers of donors. In this paper, all integers between 1 and 100 are included. Numbers greater than 100 are excluded based on findings from the study by Oudin et al. (2008), which suggest that the optimal number of donors is considerably lower. Then, for each value in this range, two loops are iterated as described below.



The outer loop will iterate over the number of donors, denoted by $m$, ranging from 1 to 100. The second (inner) loop will iterate over the 361 catchments within the GAUGED sample. At each iteration of the second loop, a catchment from the GAUGED

set is selected as the target, while the remaining 360 catchments serve as potential donors. The spatial distance between the target and each donor is calculated, and the donors are then sorted based on proximity, with the closest donor ranked first. The first $m$ closest donors are identified as the final donors. For each donor in the selected set, a set of calibrated parameters has already been obtained through the previously described donor calibration process. Utilizing these calibrated parameter sets from each of the selected donors and the forcing data from the evaluation period of the target, a simulation is conducted,

resulting in a discharge vector. This process is repeated for each of the $m$ selected donors, resulting in $m$ discharge vectors. These discharge vectors are then averaged, and the resulting average discharge is compared to the observed discharge from the evaluation period. This allows for the calculation of the KGE score. By concluding the inner loop, a total of 361 KGE scores are obtained treating each catchment in the GAUGED sample as the target catchment. The median value among these scores is saved as the ungauged performance for the $m$ considered number of donors. This inner loop process is repeated 100 times,

varying the number of donors in the range of 1 to 100. As a result, 100 median KGE scores are obtained which are shown in Fig. 12. The optimal number of donors is determined as the value that yields the highest median KGE score, which in the case of spatial proximity is found to be 50.

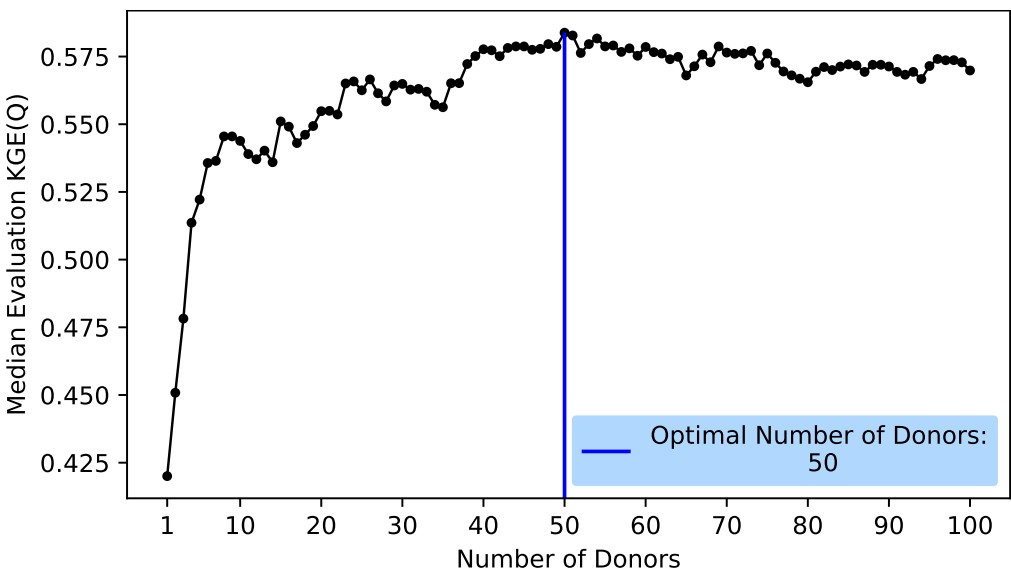

**Figure 12.** Calibrating donor count for the spatial proximity approach using the median KGE score from the evaluation subset of the GAUGED sample





## 6.2 Submethod 2: multi-attribute proximity

The multi-attribute proximity approach is similar to the spatial proximity approach, but it expands the notion of similarity
beyond spatial distance. Instead of measuring distance in a 2-dimensional spatial space, it considers a multi-dimensional space
defined by six physical and climatic attributes: area ($A$; km$^2$), median slope ($S$; %), drainage density ($DD$; %), $IP$, $IQ$, and
$T_{\min}$ (°C). These are the same attributes that are utilized in training the multi-catchment-trained LSTMs (Hashemi et al., 2022).
The Euclidean distance in this six-dimensional space is calculated using the following equation:

$$d_{\mathrm{multi-attribute}}(p,q) = \sqrt{\sum_{k=1}^{6}(p_k - q_k)^2}, \tag{3}$$

where $p$ and $q$ represent two arbitrary catchments, $p_k$ (or $q_k$) denotes the value of the k$^{\text{th}}$ attribute for catchment $p$ (or $q$), and
the index $k$ ranges from 1 to 6, corresponding to the six considered attributes.

To account for the different natures and ranges of variations among these attributes, a standardization process is applied before
using them in Equation 3 using the mean and standard deviation for each attribute (considering all catchments from both
the GAUGED and UNGAUGED samples). Regarding the $IQ$ attribute, the values used for the ungauged catchments are the
outputs obtained from the regression problem addressed in Sect. 4.

In a similar way to the spatial proximity approach, the optimal number of donors is calibrated for the multi-attribute proximity
approach (Fig. 13). In this case, it is found to be 12, which is significantly less than the optimal number of donors in the spatial
proximity approach.

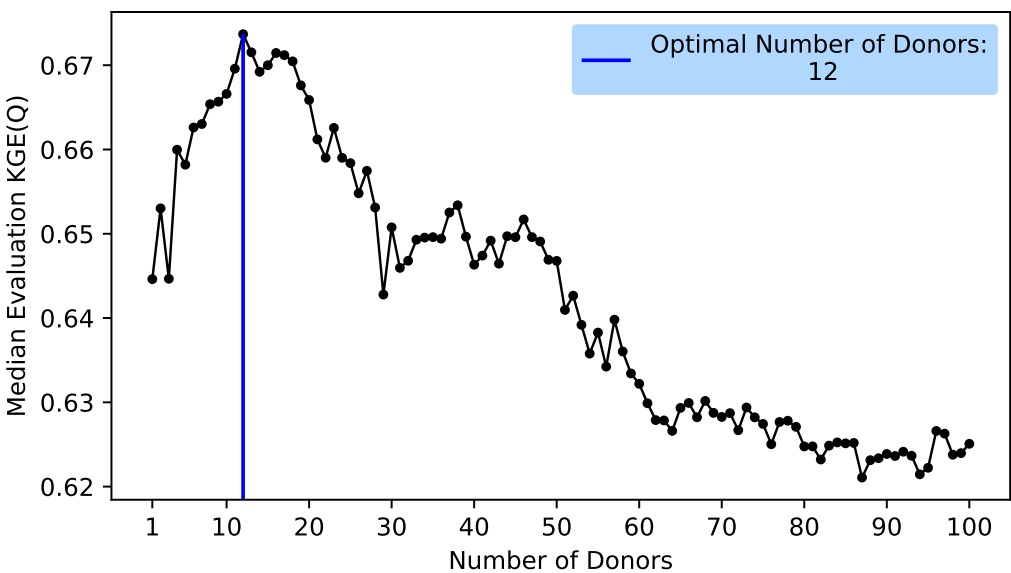

**Figure 13.** Calibrating donor count for the multi-attribute proximity approach using the median KGE score from the evaluation subset of the
GAUGED sample





### 6.3   Submethod 3: regime proximity

The third approach for donor selection involves selecting donors from the same hydrological regime as the target catchment. In this context, "regime proximity" refers to binary distances of either 0 or 1, indicating whether a donor and the target belong to the same regime. This approach differs from the previous two approaches (spatial and multi-attribute proximity) in that it directly considers the hydrological similarity without requiring distance calculations. One immediate advantage of this method is that it eliminates the need for calibrating the number of donors, which can be computationally expensive when considering

a wide range of values.

Taking into account the fully ungauged context of this study, the regimes of the targets (i.e. ungauged catchments) in this method are taken from the supervised classification described in Sect. 5.

### 6.4   Submethod 4: IQ-IP-Tmin proximity

The final donor-selection approach introduces a variation of the multi-attribute proximity method, which also shares similarities

with the regime proximity approach. Much like the multi-attribute proximity method, it measures similarity by quantifying a non-spatial distance using a set of attributes, namely the $IP$, $IQ$, and $T_{\min}$ variables, leading to its labelling as the IQ-IP-Tmin proximity method. The parallel between this variant and the regime proximity approach lies in their common utilization of the same variables to identify catchments with hydrological similarity, albeit in two different ways. The motivation behind this method lies in assessing the importance and efficacy of the $IP$, $IQ$, and $T_{\min}$ variables in two scenarios: 1. when combined

with the other attributes considered in the broader multi-attribute proximity method, and 2. when employed separately. Similar to the spatial and multi-attribute proximity methods, the calibration of donor count is performed for the IQ-IP-Tmin proximity method. The optimal number of donors is determined as six for this approach (Fig. 14), which represents a reduction by factors of eight and two compared to the optimal number of donors identified in the spatial and multi-attribute proximity methods respectively.

## 7   Results


### 7.1   PUB using LSTM

Figure 15 compares the cumulative distribution function (CDF) for the REGIME and NATIONAL LSTMs across the five regimes and within the entire UNGAUGED sample. In three regimes, notable performance differences are observed between the NATIONAL and REGIME LSTMs: Uniform, Mediterranean, and Nival. In the Uniform regime, NATIONAL achieves the

higher median KGE score of 0.41, whereas REGIME achieves a score of 0.36. With the exception of a small range of low KGE scores, the CDF curve for the NATIONAL model consistently exhibits a shift towards improved performance levels. Similarly, in the Mediterranean regime, the NATIONAL LSTM outperforms the REGIME LSTM, obtaining a median KGE score of 0.56 compared to 0.48. Also, the entire distribution of KGE scores is shifted towards better performances for the NATIONAL model when compared to the REGIME LSTM. In the Nival regime, the NATIONAL LSTM consistently outperforms the REGIME



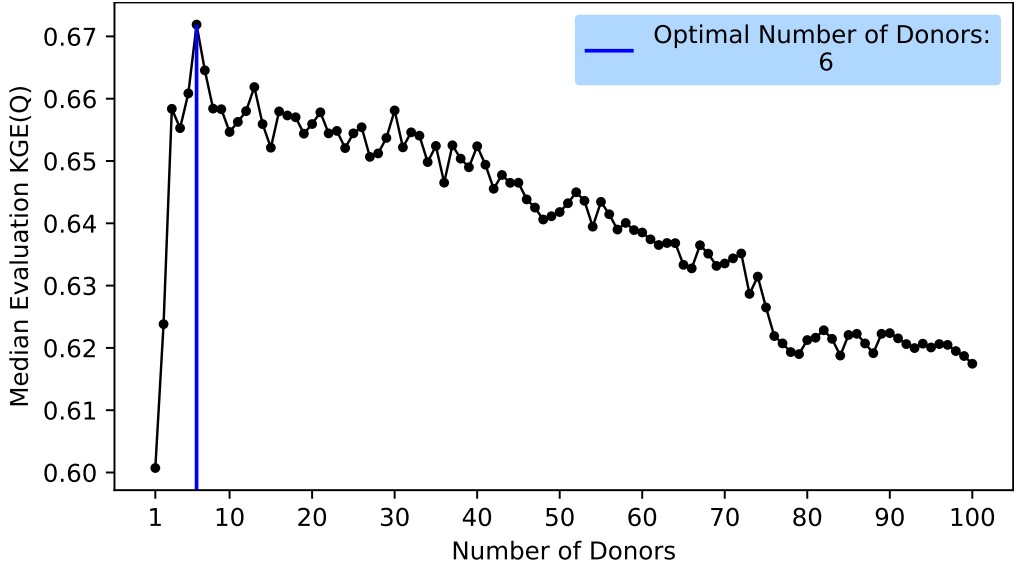

**Figure 14.** Calibrating donor count for the IQ-IP-Tmin proximity approach using the median KGE score from the evaluation subset of the GAUGED sample

LSTM, exhibiting a higher median KGE score (0.51 versus 0.47) and an important shift of the CDF curve towards improved performances across all ranges of KGE.

In the two remaining regimes, namely Nivo–Pluvial and Oceanic, the two models demonstrate close performances. In the Oceanic regimes, the CDF curves of the two models almost overlap, resulting in identical median KGE scores. Within the Nivo–Pluvial regime, the CDF curve of the NATIONAL LSTM demonstrates a noticeable shift towards better performances

for KGE scores below 0.6. However, as KGE scores increase, the CDF curves of the NATIONAL and REGIME LSTMs converge, leading to a slightly superior median KGE score for the NATIONAL model in this regime.

Considering the overall results, the NATIONAL LSTM shows superior performance in predicting runoff for the 379 catchments of the UNGAUGED sample when compared to the REGIME LSTM. This superiority is evident in both a relatively higher median KGE score (0.59 versus 0.55) and a CDF curve that is notably shifted towards better performances for intermediate KGE

scores and slightly shifted for very low and very high KGE scores.

An interesting observation is that the Oceanic and Nivo–Pluvial regimes are larger in size compared to the Uniform, Mediterranean, and Nival regimes; both within the GAUGED sample used for training and in the UNGAUGED sample (see Fig. **??**). This observation is important because it suggests that when the REGIME LSTM is trained on sufficiently large sample sizes (here, approximately 100 catchments; accounting for 0.28 of the data used in national training), it would achieve the PUB

performance of the NATIONAL model.

The PUB performance of the REGIME LSTM aligns with its performance in the gauged space (Hashemi et al., 2022): in the





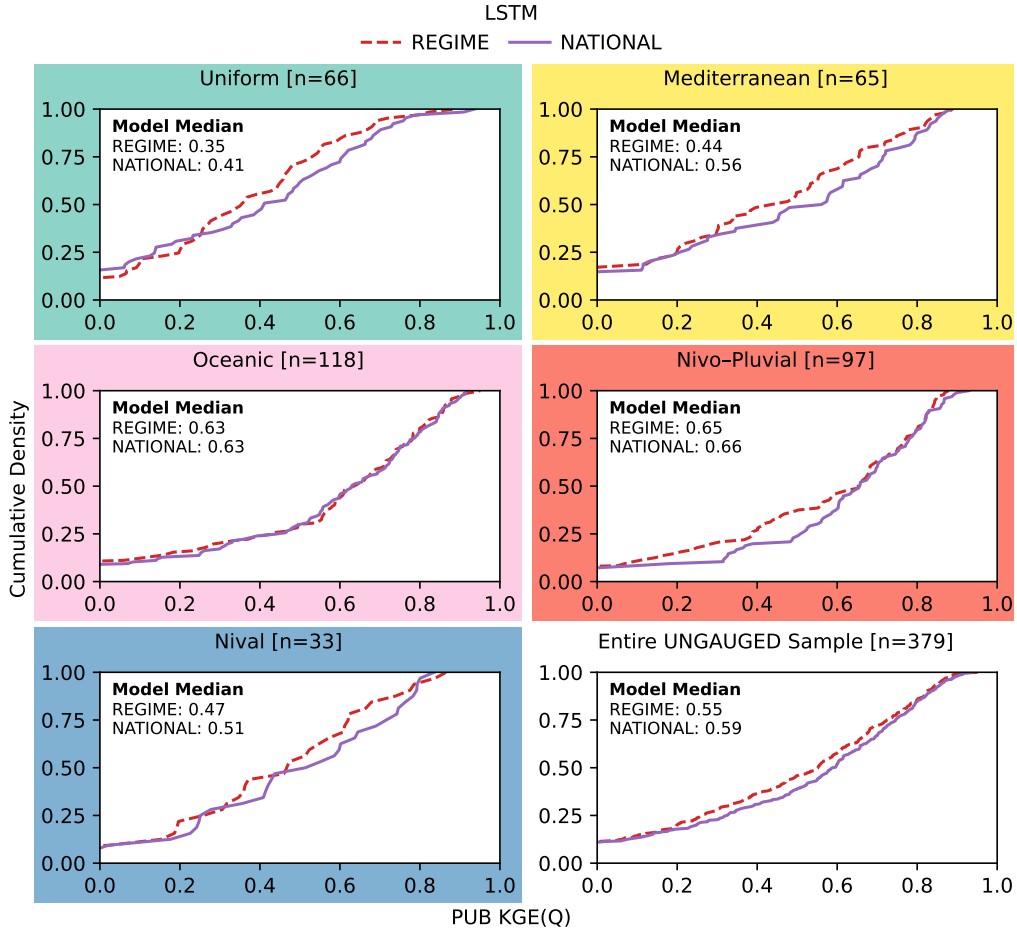

**Figure 15.** Comparison of KGE cumulative distribution functions (CDFs) for REGIME and NATIONAL LSTMs across different regimes and overall in the entire UNGAUGED sample

Uniform, Mediterranean, and Nival regimes, the NATIONAL model either shows equivalent or better performance compared to the REGIMEs, albeit in a less obvious way due to the inherently better performances in the gauged space.

## 7.2 PUB using GR4J

The CDF of KGE scores for the four GR4J regionalization approaches, namely spatial, regime, multi-attribute, and IQ-IP-Tmin proximity, are compared in Fig. 16 for the 379 catchments within the UNGAUGED sample, within each regime and overall. The four approaches show comparable performances across the Mediterranean, Nivo–Pluvial, and Nival regimes. However, in the Uniform regime, the spatial proximity approach shows significantly poorer performances across the entire range of KGE scores. Its median score stands at -0.18, contrasting with the respective medians of other methods: 0.45 (multi-attribute prox-

imity), 0.44 (IQ-IP-Tmin proximity), and 0.34 (regime proximity).





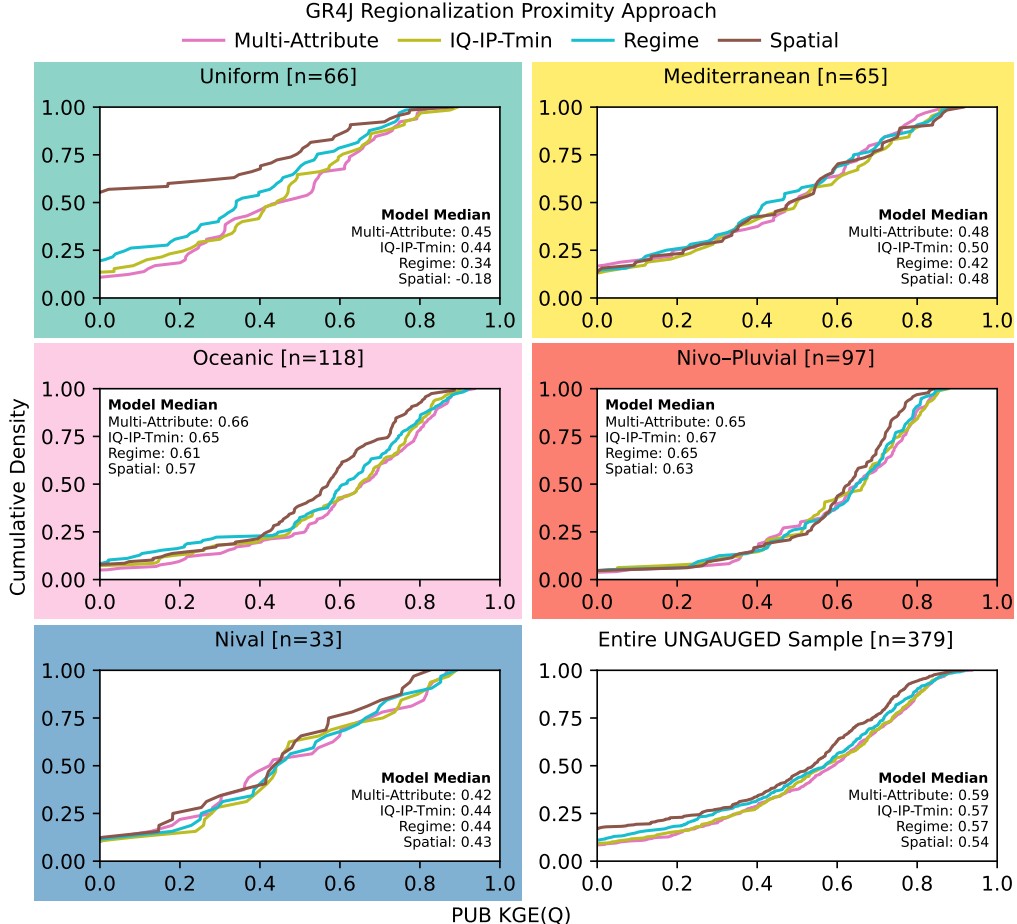

**Figure 16.** Comparing KGE cumulative distribution functions (CDFs) derived from different regionalization approaches of the GR4J model, within each regime and the entire UNGAUGED sample

Similarly, within the Oceanic regime, the spatial proximity method once again lags behind almost all other approaches across the complete range of KGE scores. However, performance difference between the spatial proximity method and other approaches is less pronounced compared to the Uniform regime. The spatial proximity method achieves a median KGE score of 0.57, which contrasts with the respective medians of the other methods: 0.66 for multi-attribute proximity, 0.65 for IQ-IP-Tmin

proximity, and 0.61 for regime proximity.

The analysis of the results from the three distance-based approaches indicates that, within the studied sample of this study, the two characteristic proximity methods (multi-attribute and IQ-IP-Tmin) provide more effective similarity guidance compared to the spatial proximity approach. This distinction becomes clearer by comparing the shape of the donor-count calibration curve and the respective value at which the global maximum is reached for each of these three methods. In the case of the

multi-attribute method (Fig. 13) and the IQ-IP-Tmin method (Fig. 14), the curves clearly reveal a global maximum at values





of 12 (multi-attribute) and 6 (IQ-IP-Tmin). These values are respectively four and eight times lower than the optimal number of donors indicated by the spatial proximity approach. In contrast, the donor-count calibration curve for the spatial proximity method takes on a monotonic shape without a distinct optimal point (Fig. 12).

The above finding contrasts with the results of the study conducted by Oudin et al. (2008), where the spatial proximity method

is found to yield the best results. There are two potential reasons that could explain this contrast. Firstly, the sample size in Oudin et al.'s 2008 study consists of 913 catchments, whereas the present study uses 361 catchments for donor-count calibration and 379 catchments for performing PUB. The spatial donor network in their study is therefore more than two times denser. Secondly, Oudin et al. (2008) do not explore the inclusion of the $IQ$, $IP$, and $T_{\min}$ attributes in their physical similarity approach.

It is observed that the IQ-IP-Tmin proximity method requires only half the number of donors compared to the multi-attribute proximity method, yet it achieves a nearly equivalent performance. This implies that among the six attributes examined in the multi-attribute approach ($IP$, $IQ$, $T_{\min}$, $A$, $S$, $DD$), the primary similarity information is conveyed by $IP$, $IQ$, and $T_{\min}$. If this were not the case, excluding the attributes of area, slope, and drainage density would likely result in a significant performance decline in the IQ-IP-Tmin proximity method, which is not observed.

The comparison of results between the multi-attribute and IQ-IP-Tmin proximity methods and the regime proximity approach reveals that the latter does not outperform the two other methods. The results suggest that, for donor identification, employing the $IP$, $IQ$, and $T_{\min}$ attributes explicitly in a distance-based form is more effective than using them for identifying hydrologically similar catchments through classification. Indeed, all three methods use the $IP$, $IQ$, and $T_{\min}$ attributes, albeit in different ways.

While the multi-attribute and IQ-IP-Tmin approaches involve explicit information from all three $IP$, $IQ$, and $T_{\min}$ variables (Equation 3), most of the regime classes are established solely based on information from one of these variables. The selection of donors in the regime proximity approach is pre-set and inflexible, determined by the assigned regime and lacking the capacity to adjust or accommodate modifications in response to intra-regime variability in the $IP$, $IQ$, and $T_{\min}$ attributes. On the contrary, in the multi-attribute and IQ-IP-Tmin proximity methods, the choice of donors is a free parameter. Within these

two approaches, which have access to the complete spectrum of these attributes, the donor calibration algorithm can explicitly target catchments with the closest $IP$, $IQ$, and $T_{\min}$. Therefore, the multi-attribute and IQ-IP-Tmin proximity methods offer a more flexible and adaptable approach to identifying donor catchments. Given the complex nature of the PUB problem and the overly simplistic regionalization approaches employed in the donor–target technique, it is plausible that the superior performance of the multi-attribute and IQ-IP-Tmin proximity methods across all regimes results from their enhanced flexibility

and increased degree of freedom.

### 7.3   LSTM and GR4J compared

The median KGE scores for both the REGIME and NATIONAL LSTMs, as well as the four GR4J proximity methods, are extracted from Fig. 15 and 16 respectively and compared in Table 7 and Fig. 17.

When comparing the best-performing LSTM with the best-performing GR4J, it is observed that in the two Mediterranean and





**Table 7.** Regime-wise comparison of median KGE scores between REGIME and NATIONAL LSTMs and the regionalized GR4J model, considering spatial, regime, multi-attribute, and IQ-IP-Tmin proximity approaches

| Catchments | LSTM | | GR4J | | | |
|---|---|---|---|---|---|---|
| | REGIME | NATIONAL | Spatial | Regime | IQ-IP-Tmin | Multi-Attribute |
| Uniform | 0.35 | 0.41 | -0.18 | 0.34 | 0.44 | 0.45 |
| Mediterranean | 0.44 | 0.56 | 0.48 | 0.42 | 0.50 | 0.48 |
| Oceanic | 0.63 | 0.63 | 0.57 | 0.61 | 0.65 | 0.66 |
| Nivo–Pluvial | 0.65 | 0.66 | 0.63 | 0.65 | 0.67 | 0.65 |
| Nival | 0.47 | 0.51 | 0.43 | 0.44 | 0.44 | 0.42 |

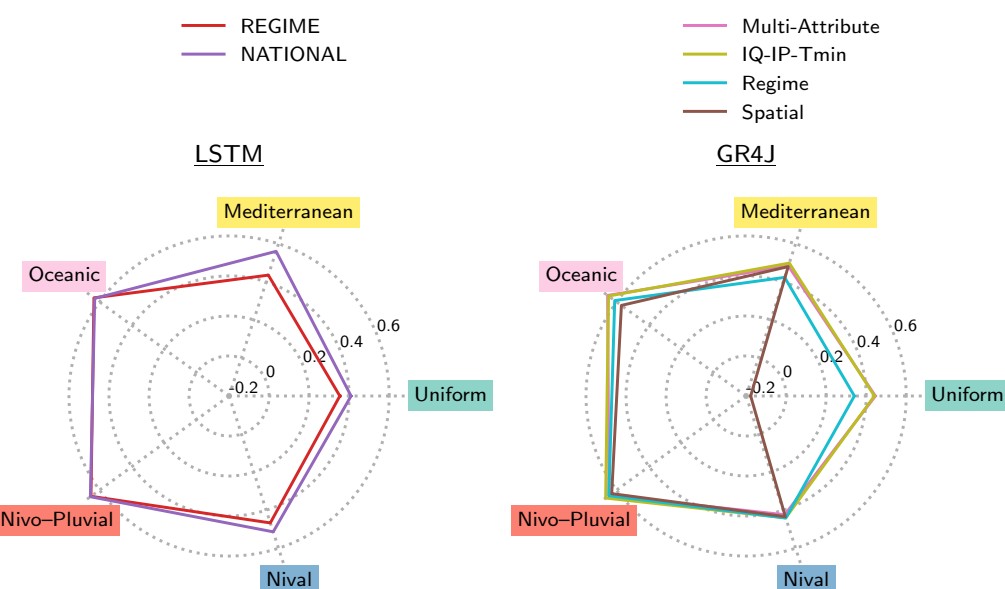

**Figure 17.** Radar chart comparing median KGE scores for REGIME and NATIONAL LSTMs with the GR4J model regionalized using spatial, regime, multi-attribute, and IQ-IP-Tmin proximity methods

Nival regimes, the LSTM significantly outperforms GR4J, with a notable difference in their median KGE scores: 0.56 versus 0.50 in the Mediterranean regime and 0.51 versus 0.44 in the Nival regime.

In the three remaining regimes, although the best-performing GR4J model leads to a better median KGE score, the difference remains insignificant; and negligible in some regimes: 0.67 versus 0.66 in the Nivo–Pluvial regime, 0.66 versus 0.63 in the Oceanic regime, and 0.45 versus 0.41 in the Uniform regime. Thus, in a regime-wise comparison, the resultant gains are higher

for LSTM compared to GR4J.

Figure 18 compares the KGE scores between the best-performing LSTM and GR4J regionalization, specifically the NA-





TIONAL LSTM and IQ-IP-Tmin proximity approach, within the Mediterranean and Nival regimes. Within each regime, three catchments are selected, labeled by numbers 1 to 3 (Fig. 18). Catchments 1 and 2 show instances where the LSTM outperforms, while catchment 3 serves as an example of superior GR4J performance. For each of these catchments, the regime of runoff (expressed as annual average monthly runoff in mm month⁻¹), the annual average daily error (computed as the difference between simulated and observed values in mm day⁻¹), and the hydrograph for one hydrologic year are provided in Fig. 19-21 (for the three catchments in the Mediterranean regime) and Fig. 22-24 (for those in the Nival regime). The trends disclosed by these figures suggest that, although the LSTM generally outperforms GR4J in the Mediterranean and Nival regimes (as shown in Table 7), a consistent pattern is not observed upon closer examination of their results. In catchments where LSTM outperforms, GR4J might have different behaviors:

– a constant overestimation, as exemplified in both the Mediterranean (the Gardon de Saint-Germain catchment; V7105210; Lozère [48][1]; Fig. 19) and Nival (the Souloise catchment; W2215030; Savoie [73]; Fig. 22) regimes;

– a continuous underestimation, as observed in the Mediterranean regime by the Golo catchment (Y7022010; Haute-Corse [2B]; Fig. 20); or

– an alternating over–under estimation, as seen in the Nival regime by the Avérole catchment (W1006010; Hautes-Alpes [05]; Fig. 23).

On the other hand, in catchments where GR4J performs better, LSTM may have:

– a constant overestimation, as evidenced in the Mediterranean regime by the Taravo catchment (Y8624010; Corse-du-Sud [2A]; Fig. 21); or

– an underestimation, as observed in the Nival regime by the Tinée catchment (Y6204020; Alpes-Maritimes [06]; Fig. 24).

## 8 Conclusions

In this paper, runoff predictions are carried out under genuine ungauged conditions for a sample of 379 catchments (the UNGAUGED sample), encompassing various hydrological regimes within France. To accomplish this, the REGIME and NATIONAL LSTMs, previously trained on a completely independent set of 361 gauged catchments (the GAUGED sample), are utilized. The LSTM predictions are benchmarked against the GR4J model, which is regionalized using four approaches: spatial proximity, multi-attribute proximity, regime proximity, and IQ-IP-Tmin proximity. Aligning with the fully ungauged context, two key steps are undertaken:

1. An XGBoost multi-class classifier is trained on the GAUGED sample to re-establish the regime classification that the REGIME LSTMs rely upon. This classification is then applied to the UNGAUGED sample.

---

[1]The numbers in the brackets represent the department numbers in which the catchments are located.





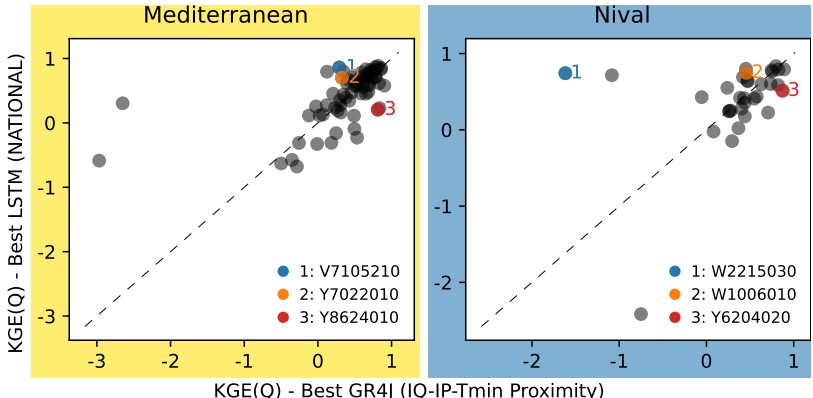

**Figure 18.** Comparison of KGE scores between the NATIONAL LSTM and IQ-IP-Tmin proximity GR4J models within the Mediterranean and Nival regimes. Each point represents a catchment. The three catchments highlighted in each regime are selected for further examination (Fig. 20-22).

2. A regime-informed neural network is trained on the GAUGED sample and is applied to the UNGAUGED sample to estimate the $IQ$ for each catchment. The $IQ$ serves as an input feature for both the REGIME and NATIONAL LSTMs and is integral to two of the regionalization methods employed for the GR4J model.

Based on the obtained results and conducted analyses, the following key conclusions are drawn:

– Despite the nearly identical gauged performances between the REGIME and NATIONAL LSTMs, the NATIONAL model shows superior ungauged performances. It outperforms in three regimes (Uniform, Mediterranean, and Nival) and also overall, while in the two other regimes (Oceanic and Nivo–Pluvial), both models achieve similar median KGE scores.

   Similar performances of the two models occur in the two regimes where the number of training catchments is much larger than in the three other regimes. These results suggest that when the data size is insufficiently large, some overfitting occurs, which does not manifest in the gauged space. Data heterogeneity prevents this at the national level by presenting a more general set of patterns, preventing the LSTM from fitting "too closely" to any one pattern (regime). Consequently, the model learns representations that are more discriminative: these characteristics become immediately apparent in the ungauged context due to the inherent need of this space for generalization ability.

– Among the four regionalization approaches tested for GR4J, the IQ-IP-Tmin proximity is found to be most least effective.

– Comparing the best performing LSTM with the best performing GR4J within each regime, in the Nival and Mediterranean regimes there is a notable outperformance of LSTM.

– The change in ungauged performance of the LSTM models showed minimal correlation with the prediction error of $IQ$ in all other regimes, with the exception of the Uniform regime. This observation aligns perfectly with the fact that the key classification variable and rule of the Uniform regime both rely on $IQ$.





Gardon de Saint-Germain at Saint-Germain-de-Calberte (V7105210; Lozère [48]; 31 km$^2$)

**Figure 19.** The Gardon de Saint-Germain catchment at Saint-Germain-de-Calberte (V7105210) in the Mediterranean regime, featuring: a. regime of runoff; b. annual-average daily error; c. hydrograph for the period 1 Aug. 1993 – 31 Jul. 1994



**Figure 20.** The Golo catchment at Omessa (Y7022010) in the Mediterranean regime, featuring: a. regime of runoff; b. annual-average daily error; c. hydrograph for the period 1 Aug. 1965 – 31 Jul. 1966





**Figure 21.** The Taravo catchment at Zigliara (Y8624010) in the Mediterranean regime, featuring: a. regime of runoff; b. annual-average daily error; c. hydrograph for the period 1 Aug. 2010 – 31 Jul. 2011







**Figure 22.** The Souloise catchment at Saint-Étienne-en-Dévoluy (W2215030) in the Nival regime, featuring: a. regime of runoff; b. annual-average daily error; c. hydrograph for the period 1 Aug. 2001 – 31 Jul. 2002



Avérole at Bessans (W1006010; Savoie [73]; 46 km$^2$)



**Figure 23.** The Avérole catchment at Bessans (W1006010) in the Nival regime, featuring: a. regime of runoff; b. annual-average daily error; c. hydrograph for the period 1 Aug. 2015 – 31 Jul. 2016





**Figure 24.** The Tinée catchment at Saint-Étienne-de-Tinée (Y6204020) in the Nival regime, featuring: a. regime of runoff; b. annual-average daily error; c. hydrograph for the period 1 Aug. 1985 – 31 Jul. 1986



*Code availability.* The repository containing the codes for the regime-informed neural network and XGBoost classification will be made available upon the paper's acceptance.

*Data availability.* The HYCAR research team at INRAE processed both the meteorological data from SAFRAN reanalysis system (Vidal et al., 2010) and the hydrometric data from SCHAPI (https://hydro.eaufrance.fr/), consolidating them into a single database named Hydro-Clim (Delaigue et al., 2020), which is used in this work.

*Author contributions.* RH conducted the experiments, wrote the first version of the manuscript, and contributed to the conceptualization of the experiments. PJ formulated the methodological framework and supervised the work. OD and SR shaped the analytical strategies and enhanced the overall clarity of the paper. All authors collaboratively engaged in defining the research questions, analyzing the results, and refining the manuscript through revisions.

*Competing interests.* The authors state that there are no competing interests known to them that might have influenced the work reported in this paper.



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
