# Peer review of "Closing the data gap: runoff prediction in fully ungauged settings using LSTM"

_Hydrology and Earth System Sciences, 2023_

## Author Comment (AC3)

*Reviewer Comment: text in black*

Author Response: text in violet

**Major:**

*[1] Abstract needs to be revised to ensure that it is concise and easy to understand enough.*

We will revise our abstract to make it more concise, easier to follow, and reflective of our study's key narrative and findings.

*[2] The methodology section needs to be shortened considerably. Piling to much content here will dilute the theme of the article. To do this, compressing current sections 3-6 into one that consists of three subsections: 3.1 predicting of IO, 3.2 Recreating regime classification, and 3.3 Conceptual benchmarking. Specifically, I recommend using only one method that works best for calculating IQ, even though multiple methods have been tested. Also, some figures and text can be placed in the attachment of the paper. The methodology section should not be overloaded with figures.*

Thank you for your constructive feedback. We are taking the following steps:

- **Compressing Sections:** We will consolidate the existing sections 3-6 into a single, streamlined section with three distinct subsections: 3.1 Predicting of IO, 3.2 Recreating Regime Classification, and 3.3 Conceptual Benchmarking.
- **Method Selection:** In line with your suggestion, we will focus on the most effective method for calculating IQ.
- **Figures and Supplementary Material:** We agree with your recommendation regarding the use of figures. To prevent the methodology section from being overloaded, we will carefully select the most impactful figures for inclusion in the main text. Additional figures and text that provide further detail but are not critical for understanding the core methodology will be moved to an attachment or supplementary material.

*[3] The discussion section is an essential part of a scientific paper, but it is missing in this article. The discussion is supposed to interpret and elucidate the significance of the study findings, justify their importance and contributions to current scientific literature, and provide specific suggestions for future research. It needs to be added in the revised manuscript.*

We thank you for your feedback on the need for a dedicated discussion section to contextualize our results within the broader scientific literature. We are committed to developing a comprehensive discussion section in the revised manuscript that will compare

our findings with those of previous studies, acknowledge differences in the overarching narrative and methodology, and explore the implications of our findings for future research.

*Minor:*

*[1] Abstract: Too many abbreviations are used in the text, which reduces the readability of the manuscript.*

Thank you for this comment. We will minimize their use and opt for self-descriptive terms to improve readability.

*[2] Lines 41-45: The importance of the PUB should not be placed in this place; it should appear at the beginning of the introduction.*

We will relocate these lines to the beginning of the introduction section as suggested.

*[3] L49: How to define the 'success' for various PUB methods?*

Thank you for this interesting question. The success of PUB methods is defined through various metrics across studies. Often, it refers to predictive ability, as expressed by metrics such as NSE (Nash-Sutcliffe Efficiency) and KGE (Kling-Gupta Efficiency). However, it can also relate to a method's generalization ability across different climatic/geological contexts, or to how adequately specific hydrological processes are reproduced by a PUB method.

*[4] Lines175-180: Overall, the all three of the authors' stated contributions are a bit of stretch. Firstly, similar studies have not been performed within the French context cannot be as a novelty of this research. Also, the third contribution needs to be revisited as previous studies (searching by google scholar) have compared the performance of LSTM and GR4J models in runoff simulations.*

Thank you for your comment. We acknowledge that the main novelty aspect of our paper, which is detailed above, may not have been clearly articulated in the current manuscript version. In alignment with HESS guidelines (as encountered during the submission process), we have highlighted one novelty of our research within the French context under the 'New Data Set' metric as we are not aware of any PUB studies utilizing LSTMs within the French context with the SAFRAN dataset we employed.

Regarding the comparison between GR4J and LSTM models in runoff simulations, we acknowledge the existence of the study by Georgy Ayzel et al (https://doi.org/10.1051/e3sconf/202016301001). Our study's overarching narrative and methodology differ from those of the cited paper. Ayzel et al.'s research employs k-fold splitting for both the training-test of the LSTM and the calibration-validation of GR4J, conducted exclusively in ungauged spaces. Conversely, our approach involves the training and calibration of LSTM and GR4J models in the gauged space using a training set, with performance loss measured on a different dataset (test set) in the ungauged space. We will further clarify these distinctions in the revised manuscript to better articulate the unique contributions and novelty of our research in response to your comment.

[5] *Figure 3, the legend and symbol colors in Figure 3f need to be redrawn.*

*We will make the necessary adjustments to the legend and symbol colors in Figure 3f as suggested.*

[6] *Figure 5: It is recommended that Figure 5 be deleted, as similar information is already included in Table 2.*

We included Figure 5 for its utility in facilitating a rapid visual comparison. We acknowledge your point regarding its redundancy with Table 2. We will remove Figure 5 as recommended.

[7] *The hydrologic regime classification should be placed where it first appears, i.e. in the caption of Figure 1.*

Thank you for your comment. It will be relocated to the caption of Figure 1 as suggested.

[8] *Briefly describe the four parameter regionalisation methods used, and Figures 12-14 should be moved to the supplementary material.*

A concise description of the four parameter regionalization methods will be added to the manuscript. Additionally, Figures 12-14 will be moved to the supplementary material in line with your recommendation.

[9] *It makes no sense to compare the performance of the best-performing LSTM and GR4J in three typical basins, except to further increase the length of the article. Thus, please remove Figure 19-24 and corresponding description in the paper.*

We included Figures 19-24 to illustrate specific instances where the performance of the LSTM and GR4J models significantly diverges, sometimes with one model completely misrepresenting a typical hydrological regime. We acknowledge your concern regarding article length. Understanding that these sections serve primarily as illustrative examples, we will remove them to streamline the manuscript.

*[10] Conclusion only summarizes the main findings, and results that are not relevant to the research objectives need to be removed, such as the content in lines 576-582.*

We will refine the conclusion to more explicitly emphasize the main findings in direct relation to the study's objectives, ensuring a much closer alignment and removing content not relevant to these objectives.